# Energy Guided Diffusion for Generating Neurally Exciting Images

**Paweł A. Pierzchlewicz**[*,1-2], **Konstantin F. Willeke**[1-2], **Arne F. Nix**[1-2], **Pavithra Elumalai**[2],
**Kelli Restivo**[3-4], **Tori Shinn**[3-4], **Cate Nealley**[3-4], **Gabrielle Rodriguez**[3-4],
**Saumil Patel**[3-4], **Katrin Franke**[3-4], **Andreas S. Tolias**[3-5], **Fabian H. Sinz**[1-4]

[1]Institute for Bioinformatics and Medical Informatics, Tübingen University, Tübingen, Germany
[2]Institute of Computer Science and Campus Institute Data Science, University of Göttingen, Germany
[3]Department of Neuroscience, Baylor College of Medicine, Houston, TX, USA
[4]Center for Neuroscience and Artificial Intelligence, Baylor College of Medicine, Houston, TX, USA
[5]Department of Electrical and Computer Engineering, Rice University, Houston, TX, USA
[*]`ppierzc@cs.uni-goettingen.de`

## Abstract

In recent years, most exciting inputs (MEIs) synthesized from encoding models of neuronal activity have become an established method for studying tuning properties of biological and artificial visual systems. However, as we move up the visual hierarchy, the complexity of neuronal computations increases. Consequently, it becomes more challenging to model neuronal activity, requiring more complex models. In this study, we introduce a novel readout architecture inspired by the mechanism of visual attention. This new architecture, which we call attention readout, together with a data-driven convolutional core outperforms previous task-driven models in predicting the activity of neurons in macaque area V4. However, as our predictive network becomes deeper and more complex, synthesizing MEIs via straightforward gradient ascent (GA) can struggle to produce qualitatively good results and overfit to idiosyncrasies of a more complex model, potentially decreasing the MEI's model-to-brain transferability. To solve this problem, we propose a diffusion-based method for generating MEIs via Energy Guidance (EGG). We show that for models of macaque V4, EGG generates single neuron MEIs that generalize better across varying model architectures than the state-of-the-art GA, while at the same time reducing computational costs by a factor of 4.7x, facilitating experimentally challenging closed-loop experiments. Furthermore, EGG diffusion can be used to generate other neurally exciting images, like most exciting naturalistic images that are on par with a selection of highly activating natural images, or image reconstructions that generalize better across architectures. Finally, EGG is simple to implement, requires no retraining of the diffusion model, and can easily be generalized to provide other characterizations of the visual system, such as invariances. Thus, EGG provides a general and flexible framework to study the coding properties of the visual system in the context of natural images.[1]

## 1 Introduction

From the early works of Hubel and Wiesel [1], visual neuroscience has used the preferred stimuli of visual neurons to gain insight into the information processing in the brain. In recent years, deep learning has made big strides in predicting neuronal responses [2–16] enabling *in silico* stimulus

---

[1]The code is available at `https://github.com/sinzlab/energy-guided-diffusion`

37th Conference on Neural Information Processing Systems (NeurIPS 2023).

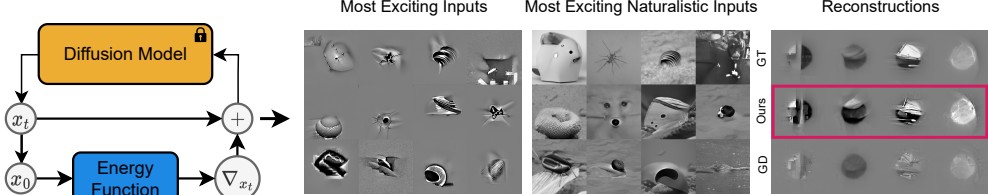

Figure 1: **Schematic** of the EGG diffusion method with a pre-trained diffusion model. Examples of applications: **Left**: Most Exciting Inputs for different neurons, **Middle:** Most Exciting Naturalistic Inputs matched unit-wise to the MEIs. **Right**: Reconstructions in comparison to the ground truth (top) and gradient descent optimized (bottom).

synthesis of non-parametric most exciting inputs (MEIs) [17–19]. MEIs are images that strongly drive a selected neuron and can thus provide insights into its tuning properties. Up until now, they have been successfully used to find novel properties of neurons in various brain areas in mice and macaques [17–24].

However, as we move up the visual hierarchy, such as monkey visual area V4 and IT, the increasing non-linearity of neuronal responses with respect to the visual stimulus makes it more challenging to ❶ obtain models with high predictive performance for single neurons, and ❷ optimize perceptually plausible MEIs, that is, those not corrupted by adversarial high-frequency noise for example. Particularly, area V4 is known to be influenced by attention effects [25], and shifts in attention before the onset of saccades can change the location of its neurons' receptive fields [26, 27]. When models become more complex or units are taken from deeper layers of a network, existing MEI optimization methods based on gradient ascent (GA) can sometimes have difficulties producing qualitatively good results [28] and can overfit to the idiosyncrasies of more complex models, potentially decreasing the MEI's model-to-brain transferability. Typically, these challenges are addressed by biasing MEIs towards the statistic of natural images, for instance by gradient pre-conditioning [28], by including a total variation loss to reduce high-frequency noise [29] or by image synthesis via GANs [19]. However, as discussed by Engstrom et al. [30] and Feather et al. [31] including additional priors into the generation process can result in obfuscated model biases.

Here, we make two contributions towards the above points: ❶ We introduce a new model architecture, called the attention readout, for predicting the activity of neurons in macaque area V4 , which together with a data-driven convolutional core outperforms previous task-driven models [24, 32]. ❷ To improve the quality of MEI synthesis we introduce a novel method for optimizing MEIs via Energy Guided Diffusion (EGG). EGG diffusion guides a pre-trained diffusion model with a learned neuronal encoding model to generate MEIs with a bias towards natural image statistics. Our proposed EGG method is simple to implement and, in contrast to similar approaches [33–35], requires no retraining of the diffusion model (Fig. 1). We show that EGG diffusion not only yields MEIs that generalize better across architectures and are thus expected to drive real neurons equally well or better than GA-based MEIs but also provides a significant (4.7x) speed up over the standard GA method enhancing its utility for close-loop experiments such as inception loops [17, 18, 20, 24]. Since optimizing MEIs for thousands of neurons can take weeks [24], such a speed-up directly decreases the energy footprint of this technique. Moreover, the rapid verification of synthesized images *in vivo* is particularly important for close-loop experiments given that maintaining the stability of single unit recordings is challenging, and there's also the issue of representational drift [36], where tuning functions can change over time. We also demonstrate that EGG diffusion straightforwardly generalizes to provide other characterizations of the visual system that can be phrased as an inverse problem, such as image reconstructions based on neuronal responses. The flexibility and generality of EGG thus make it a powerful tool for investigating the neural mechanisms underlying visual processing.

## 2 Attention readout for macaque area V4

**Background**  Deep network-based encoding models have set new standards in predicting neuronal responses to natural images [2–15]. Virtually all architectures of these encoding models consist of at least two parts: a *core* and a *readout*. The core is usually implemented via a convolutional

network that extracts non-linear features $\Phi(\boldsymbol{x})$ from the visual input and is shared across all neurons to be predicted. It is usually trained through one of two paradigms: i) *task-driven*, where the core is pre-trained on a different task like object recognition [3, 4, 37–39] and then only the readout is trained to predict the neurons' responses or ii) *data-driven* where the model is trained end-to-end to predict the neurons' responses. The *readout* is a collection of predictors that map the core's features to responses of individual neurons. With a few exceptions [40], the readout components and its parameters are neuron-specific and are therefore kept simple. Typically, the readout is implemented by a linear layer with a rectifying non-linearity. Different readouts differ by the constraints they put on the linear layer to reduce the number of parameters [3, 4, 37, 40–42]. One key assumption all current readout designs make is that the readout mechanism does not change with the stimulus. In particular, this means that the location of the receptive field is fixed. While this assumption is reasonable for early visual areas like V1, it is not necessarily true for higher or mid-level areas such as macaque V4, which are known to be affected by attention effects and can even shift the location of the receptive fields [26]. This motivated us to create a more flexible readout mechanism for V4.

**State-of-the-art model: Robust ResNet core with Gaussian readout**    In this study, we compare our data-driven model to a task-driven model [24], which is also composed of a *core* and *readout*. The core is a pre-trained robust ResNet50 ($L_2, \varepsilon = 0.1$) [43, 44]. We use the layers up to layer 3 in the ResNet, which has 1,024 channels, thus providing a 1,024 dimension feature space. Then batch normalization is applied [45], followed by a ReLU non-linearity. The *Gaussian readout* [40] learns the position of each neuron and extracts a feature vector at this position. During training, the positions are sampled from a 2D Gaussian distribution with means $\mu_n$ and $\Sigma_n$, during inference the $\mu_n$ positions are used. Then the extracted features are used in a linear non-linear model to predict neuronal responses. We will refer to this model as the **Gaussian model**.

**Proposed model: Data-driven core with attention readout**    The predictive model is trained from scratch to predict the neuronal responses in an end-to-end fashion. Following Lurz et al. [40], the architecture is comprised of two main components. First, the *core*, a four-layer CNN with 64 channels per layer with an architecture identical to Lurz et al. [40]. Secondly, the attention *readout*, which builds upon the attention mechanism [46, 47] as it is used in the popular transformer architecture [48]. After adding a fixed positional embedding to $\Phi(\boldsymbol{x})$ and normalization through LayerNorm [49] to get $\tilde{\Phi}(\boldsymbol{x})$, key and value embeddings are extracted from the core representation. This is done by position-wise linear projections $V \in \mathbb{R}^{c \times d_k}$ and $U \in \mathbb{R}^{c \times d_v}$ both of which have parameters shared across all neurons. Then, for each neuron a learned query vector $\mathbf{q}_n \in \mathbb{R}^{d_k}$ is compared with each position's key embedding using scaled dot-product attention [48].

$$\alpha_n = \mathrm{softmax} \left( \sum_{c,d_k} \frac{\tilde{\Phi}(\boldsymbol{x})_c W_{c,d_k} q_{n,d_k}}{\sqrt{d_k}} \right) \qquad (1)$$

The result is a spatially normalized attention map $\alpha_n \in \mathbb{R}^{h \times w \times 1}$ that indicates the most important feature locations for a neuron $n$ given an input image. Using this attention map to compute a weighted sum of the value embeddings gives us a single feature vector for each neuron. Finally, a neuron-specific affine projection with ELU non-linearity [50] gives rise to the predicted spike rate $\hat{r}_n$ (Fig. 2A). The model training is performed by minimizing the Poisson loss using the same setup as described in Willeke et al. [24]. We will refer to this model as the **Attention model**.

**Training data**    We use data from 1,244 Macaque V4 neurons from Willeke et al. [24] and briefly summarize their data acquisition in the supplementary materials section A.1.

**Results**    Our Attention model significantly outperforms the Gaussian model in predicting neuronal responses of macaque V4 cells on unseen natural and model-derived images. We evaluate the model performance by the correlation between the model's prediction and the averages of actual neuron responses across multiple presentations of a set of test images, as described by Willeke et al. [24]. We compared this predictive performance to the Gaussian model [44] on 1,244 individual neurons (Fig. 2B). The Attention model significantly outperforms the Gaussian model by 12% (Wilcoxon signed-rank test, p-value $= 6.79 \cdot 10^{-82}$). In addition, we evaluated the new readout on how well it predicts the real neuronal responses to 48 MEIs generated from the Gaussian model [see 24] and 7 control natural images. Our Attention model is better at predicting real neuronal responses,

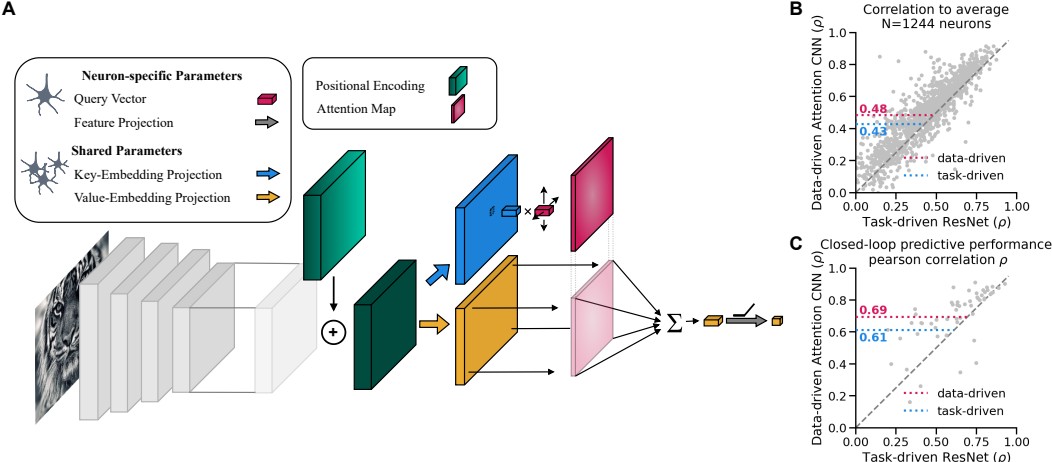

Figure 2: **a**) Schematic of the Attention Readout. **b**) Correlation to average scores for 1,244 neurons. The Attention model (pink) shows a significant (as per the Wilcoxon signed rank test, p-value $= 6.79 \cdot 10^{-82}$) increase in the mean correlation to average in comparison to the Gaussian model (blue). **c**) Predictive performance comparison of the two models in a closed-loop MEI evaluation setting. Showing that the data-driven with attention readout model better predicts the in-vivo responses of the MEIs.

even for MEIs of another architecture (Fig. 2C). Please note that Willeke et al. [24] experimentally verified MEIs in only a subset of neurons and only used the neurons with high functional consistency across different experimental sessions. For that reason, we too can only compare the performance of model-derived MEIs on this subset of neurons. We additionally show that the Attention model and Gaussian model show representational similarity (see Table S1) and that the Attention model uses its ability to shift its receptive field (Fig. S1).

| Readout \ Core | Task-Driven | Data-Driven |
|---|---|---|
| Factorized | - | 0.153 |
| Gaussian | 0.262 | 0.229 |
| Attention | 0.276 | **0.294** |

Table 1: Ablation study test correlation comparison for combinations of different cores and readouts. Bold indicates the best-performing model.

**Ablation Study** We perform an ablation study comparing the effects of the choice of core and readout on the performance in terms of test correlation (Table 1). We identify that the data-driven core + Attention readout model outperforms all previous setups. Furthermore, the ablation study shows that the Attention readout generally improves performance across cores.

## 3 Energy guided diffusion (EGG)

### 3.1 Algorithm and methods

In this section, we describe our approach to extract tuning properties of neuronal encoding models using a natural image prior as described by a diffusion model. In brief, we use previously established links between diffusion and score-based models and the fact that many tuning properties can be described as inverse problems (most exciting image, image reconstruction from neuronal activity, etc.) to combine an energy landscape defined by the neuronal encoding model with the energy landscape defined by the diffusion model and synthesize images via energy minimization. We show that this method leads to better generalization of MEIs and image reconstructions across architectures, faster generation, and allows for generating natural-looking stimuli.

**Background: diffusion models** Recently, Denoising Diffusion Probabilistic Models (DDPMs) have proved to be successful at generating high-quality images [33, 51–56]. These models can be formalized as a variational autoencoder with a fixed encoder $x_0 \mapsto x_T$ that turns a clean sample $x_0$ into a noisy one $x_T$ by repeated addition of Gaussian noise, and a learned decoder $x_T \mapsto x_0$ [33],

which is often described as inverting a diffusion process [51]. After training, the sampling process is initialized with a standard Normal sample $\boldsymbol{x}_T \sim \mathcal{N}(\boldsymbol{0}, \boldsymbol{I})$ which is iteratively "denoised" for $T$ steps until $\boldsymbol{x}_0$ is reached. In the encoding, each step $t$ corresponds to a particular noise level such that

$$\boldsymbol{x}_t = \sqrt{\bar{\alpha}_t}\boldsymbol{x}_0 + \sqrt{1 - \bar{\alpha}_t}\boldsymbol{\varepsilon}_0 \tag{2}$$

where $\bar{\alpha}_t$ controls the signal strength at time $t$ and $\boldsymbol{\varepsilon}_0 \sim \mathcal{N}(\boldsymbol{0}, \boldsymbol{I})$ is independent Gaussian noise. In the decoding step, the diffusion model predicts the noise component $\varepsilon_\theta(\boldsymbol{x}_t, t)$ at each step $t$ of the diffusion process [33]. Then the sampling is performed according to

$$\boldsymbol{x}_{t-1} = \frac{1}{\sqrt{\alpha_t}} \left( \boldsymbol{x}_t - \frac{1 - \alpha_t}{\sqrt{1 - \bar{\alpha}_t}} \varepsilon_\theta\left(\boldsymbol{x}_t, t\right) \right) + \sigma_t \boldsymbol{z} \tag{3}$$

where $\boldsymbol{z} \sim \mathcal{N}(\boldsymbol{0}, \boldsymbol{I})$.

Several previous works have established a link between diffusion models and energy-based models [57–59]. In particular, the diffusion model $\varepsilon_\theta\left(\boldsymbol{x}_t, t\right)$ can be interpreted as a *score function*, i.e. the gradient of a log-density or energy w.r.t. the data $\nabla_{\boldsymbol{x}} \log p(\boldsymbol{x})$ [60]. This link is particularly useful since combining two density models via a product is equivalent to adding their score functions.

**EnerGy Guided Diffusion (EGG)**     To optimize neurally exciting images, we require a method that can guide diffusion models via neural encoding models. The parameterization of diffusion models introduced by Ho et al. [33] only allows for the unconditioned generation of samples. Dhariwal and Nichol [53] introduced a method for sampling from a conditional distribution $p_t(\boldsymbol{x} \mid \boldsymbol{y})$, with diffusion models using a classifier $p_t(\boldsymbol{y} \mid \boldsymbol{x})$ known as classifier guidance. However, this method requires i) the classifier to be trained on the noisy images, and ii) is limited to conditions for which classification makes sense. Essentially, this method relies on computing the score of the posterior distribution.

$$\nabla_{\boldsymbol{x}_t} \log p(\boldsymbol{x}_t \mid \boldsymbol{y}) = \nabla_{\boldsymbol{x}_t} \log p(\boldsymbol{x}_t) + \nabla_{\boldsymbol{x}_t} \log p(\boldsymbol{y} \mid \boldsymbol{x}_t) \tag{4}$$

For classifier-guidance, the gradient of a model $\nabla_{\boldsymbol{x}_t} \log p(\boldsymbol{y} \mid \boldsymbol{x}_t)$ with respect to the noisy input $\boldsymbol{x}_t$ is combined with the diffusion model $\nabla_{\boldsymbol{x}_t} \log p(\boldsymbol{x}_t)$, resulting in samples $\boldsymbol{x}_0$ conditioned on the class $\boldsymbol{y}$. Note that this requires a model $\nabla_{\boldsymbol{x}_t} \log p(\boldsymbol{y} \mid \boldsymbol{x}_t)$ that has been trained on noisy samples of the diffusion before. Here we extend this approach to i) use neuronal encoding models, such as the ones described above, to guide the diffusion process and ii) use a model trained on *clean* samples only. We achieve i) by defining conditioning as a sum of energies. Specifically, we redefine equation (4) in terms of the output of the diffusion model $\varepsilon_\theta(\boldsymbol{x}_t, t)$ and an arbitrary energy function $E(\boldsymbol{x}_t, t)$:

$$\bar{\varepsilon}(\boldsymbol{x}_t, t) = \varepsilon_\theta(\boldsymbol{x}_t, t) + \lambda_t \nabla_{\boldsymbol{x}_t} E(\boldsymbol{x}_t, t) \tag{5}$$

where $\lambda_t$ is the energy scale. This takes advantage of the fact that sampling in DDPMs is functionally equivalent to Langevin dynamics [51]. Langevin dynamics generally define the movement of particles in an energy field and in the special case when $E(x) = -\log p(x)$, Langevin dynamics generates samples from $p(x)$. For this study, we use a constant value of $\lambda$ and normalize the gradient of the energy function to a magnitude of 1.

To achieve ii) we use an approximate clean sample $\bar{\boldsymbol{x}}_0$, i.e. the original image, that can be estimated at each time step $t$. This is achieved by a simple trick introduced in Li et al. [61]. By inverting the forward diffusion process, with the assumption that the predicted $\varepsilon_\theta(\boldsymbol{x}_t, t)$ is the true noise:

$$\bar{\boldsymbol{x}}_0(\boldsymbol{x}_t, t) = \frac{1}{\sqrt{\bar{\alpha}_t}}(\boldsymbol{x}_t - \sqrt{1 - \bar{\alpha}_t}\varepsilon_\theta(\boldsymbol{x}_t, t)). \tag{6}$$

As a result, the energy function receives inputs that are in the domain of $\boldsymbol{x}_0$ at much earlier time steps $t$, and hence makes it feasible to use energy functions only defined on $\boldsymbol{x}_0$ and not $\boldsymbol{x}_t$, dropping the requirement to provide an energy $E(\boldsymbol{x}_t, t)$ that can take noisy images. Thus, the new score can be defined as

$$\bar{\varepsilon}(\boldsymbol{x}_t, t) = \varepsilon_\theta(\boldsymbol{x}_t, t) + \lambda_t \nabla_{\boldsymbol{x}_t} E(\bar{\boldsymbol{x}}_0(\boldsymbol{x}_t, t)) \tag{7}$$

This is particularly relevant in the domain of neural system identification, as encoding models are trained on neuronal responses to natural "clean" images [2–15, 17, 21, 24, 40]. To get an energy that can understand noisy images would require showing the noisy images to the animals in experiments, which would make the use of this method prohibitively more difficult. Therefore, a guidance method that does not require training an additional model on noisy images allows researchers to apply EGG diffusion directly to existing models trained on neuronal responses and extract tuning properties from them.

**Related work**  Many other methods have been proposed to condition the samples of diffusion processes on additional information. Ho and Salimans [55] provided a method that addressed the second requirement of classifier-guidance by incorporating the condition $y$ into the denoiser $\varepsilon_\theta(x_t, t, y)$. However, to introduce a conditioning domain $y$ in this classifier-free guidance, the whole diffusion model needs to be retrained. Furthermore, this link between diffusion models and energy-based models allowed several previous works to compose diffusion models to generate outputs that contain multiple desired aspects of a generated image [57–59]. However, these studies focus solely on generalizing the classifier-free guidance to allow guiding diffusion models with other diffusion models. Nichol and Dhariwal [52] have used a similar gradient conditioning to guide the diffusion process using the gradient of the dot product of the CLIP image and text vectors. It has been shown that CLIP models that have not been trained on noisy images can be used for guiding diffusion models [62, 63]. Kadkhodaie and Simoncelli [64] introduced a stochastic coarse-to-fine gradient ascent procedure for generating samples from the implicit prior embedded within a CNN. While we were working on this project, Feng et al. [65] published a preprint where they used the score-based definition of diffusion models to introduce an image-based prior for inverse problems where the posterior score function is available. This work is most closely related to our approach. However, they focus on how to obtain samples and likelihoods from the true posterior. For that reason, they need guiding models to be proper score functions. We do not need that constraint and focus on guiding inverse problems defined by a more general energy function and focus particularly on the application to neuronal encoding models.

**Image preprocessing for neural models**  The neural models used in this study expect $100 \times 100$ images in grayscale. However, the output of the ImageNet pre-trained Ablated Diffusion Model (ADM) [53] is a $256 \times 256$ RGB image. We, therefore, use an additional compatibility step that performs i) downsampling from $256 \times 256 \rightarrow 100 \times 100$ with bilinear interpolation and ii) takes the mean across color channels providing the grayscale image. Each of these preprocessing steps is differentiable and is thus used end-to-end when generating the image.

## 3.2  Experiments

**Most exciting images**  We apply EGG diffusion to characterize the properties of neurons in macaque area V4. For each of these experiments, we use the pre-trained ADM diffusion model trained on $256 \times 256$ ImageNet images from Dhariwal and Nichol [53]. In each of our experiments, we consider two paradigms: 1) **within** architecture, where we use two independently pre-trained ensembles containing 5 models of the same architecture (Gaussian model or Attention model). We generate images on one and evaluate them on the other. 2) **cross** architecture, two independently pre-trained ensembles containing 5 models of different architectures (Gaussian model and Attention model). We demonstrate EGG on three tasks ❶ Most Exciting Input (MEI) generation, where the generation method needs to generate an image that maximally excites an individual neuron, ❷ naturalistic image generation, where a natural-looking image is generated that maximizes individual neuron responses, and ❸ reconstruction of the input image from predicted neuronal responses. Running the experiments required a total of 7 GPU days. All computations were performed on a single consumer-grade GPU: NVIDIA GeForce RTX 3090 or NVIDIA GeForce RTX 2080 Ti depending on the availability.

MEIs have served as a powerful tool for visualizing features of a network, providing insights and testable predictions [17–21, 23, 66]. For the generation of MEIs, we selected 90 units at random from a subset of all 1,244 for which both the Gaussian model and the Attention model achieve at least a correlation of 0.5 to the average responses across repeated presentations. We compare our method to a vanilla gradient ascent (GA) method [24] which optimizes the pixels of an input image $x$ to obtain the maximal response of the selected neuron. For the GA method, we use Gaussian blur preconditioning of the gradient. The stochastic gradient descent (SGD) optimizer was used with a learning rate of 10 and the image was optimized for 1,000 steps. We also evaluated other setups for the GA method without finding major differences (see Fig. S2). We define EGG diffusion with the energy function $E(\bar{x}_0) = f_i(\bar{x}_0)$, where $f_i$ is the $i$-th neuron model and $\bar{x}_0$ is the estimated clean sample. We optimize MEIs for both the Gaussian model and the Attention model. We set the energy scale to $\lambda = 10$ for the Gaussian model and $\lambda = 5$ for the Attention model. $\lambda$ was chosen via a grid search, for more details refer to Fig. 5B. The diffusion process was run for 100 respaced time steps for the Gaussian model and 50 respaced time steps for the Attention model. For both EGG and GA, we set the norm of the $100 \times 100$ image to a fixed value of 25. For each of the methods, we chose

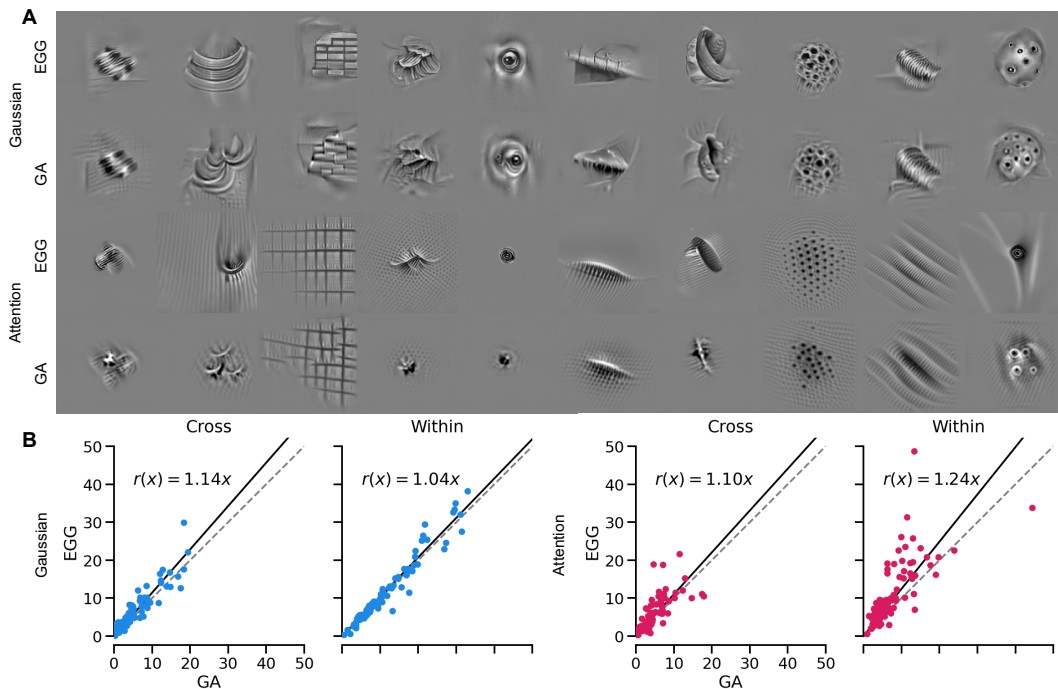

Figure 3: **a)** Examples of MEIs optimized using EGG diffusion and GA for macaque V4 Gaussian and Attention models. **b)** Comparison of activations for different neurons between EGG diffusion and GA on the Within and Cross Architecture validation paradigms. Line fits obtained via Huber regression with $\varepsilon = 1.1$. Curated image selection to show various properties of the neurons like fur, eyes, curves and edges.

the best of 3 MEIs optimized from different seeds. We show the influence of the initial seed on the generated MEI in figure S3. Furthermore, the images that are generated by the ADM model are RGB. We show examples of the color outputs in figure S4.

We show some examples of MEIs generated with EGG diffusion and GA for the two architectures in figure 3A. For more examples, refer to the supplementary materials figure S5. We find that the EGG-generated MEIs are significantly better (Attention) or similarly (Gaussian) activating within architectures and are significantly better at generalizing across architectures (Fig. 3B). This can also be observed by a significant increase in the mean activation across all units (Table 2). Perceptually, EGG-generated MEIs of the Attention model looked more complex and natural than the GA-generated MEIs, and more similar to MEIs of the Gaussian model pre-trained on natural image classification.

Comparing EGG-based MEIs to the ones found by Willeke et al. [24] using GA, we find that the preferred image feature is usually preserved, but MEIs generated for the Attention model are in most cases smaller in visual angle than their Gaussian model counterparts (Fig. S5). To quantify that the MEIs from the Attention model are smaller we compute an isotropic Gaussian envelope for the MEIs. We find that the Attention model generates MEIs for which their Gaussian envelope on average is smaller than for the Gaussian MEIs ($\sigma_{At} = 49.62$ vs $\sigma_{Ga} = 55.36$, Wilcoxon signed rank test p-value: 0.0078).

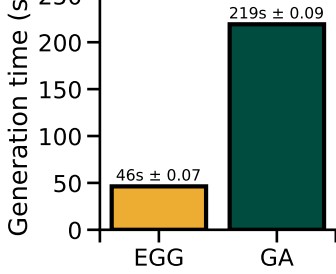

Figure 4: Mean comparison of the generation times between the EGG and GA (error bars denote standard error).

Finally, EGG diffusion is almost 4.7-fold faster than GA, requiring only on average 46s per MEI in comparison to the required 219s for the GA method (Fig. 4) on a single NVIDIA GeForce RTX 3090 across 10 repetitions. This

| Paradigm | Within (Gaussian) | Cross (Gaussian) | Within (Attention) | Cross (Attention) |
|---|---|---|---|---|
| Gradient Ascent | 11.43 | 5.51 | 7.59 | 4.42 |
| EGG Diffusion | **11.76** | **6.53**$^\dagger$ | **10.56**$^\dagger$ | **5.50**$^\dagger$ |

Table 2: Comparison of the average unit activations in response to MEIs in two paradigms 1) within architectures and 2) cross architectures, for two architectures Gaussian and Attention. Bold marks the method which has higher mean activation, and the $\dagger$ marks the increases which are statistically significant (Wilcoxon signed-rank test, respective p-values: $0.08, 2.87 \cdot 10^{-6}, 2.84 \cdot 10^{-10}, 4.39 \cdot 10^{-5}$). The architecture in the bracket indicates the generator architecture.

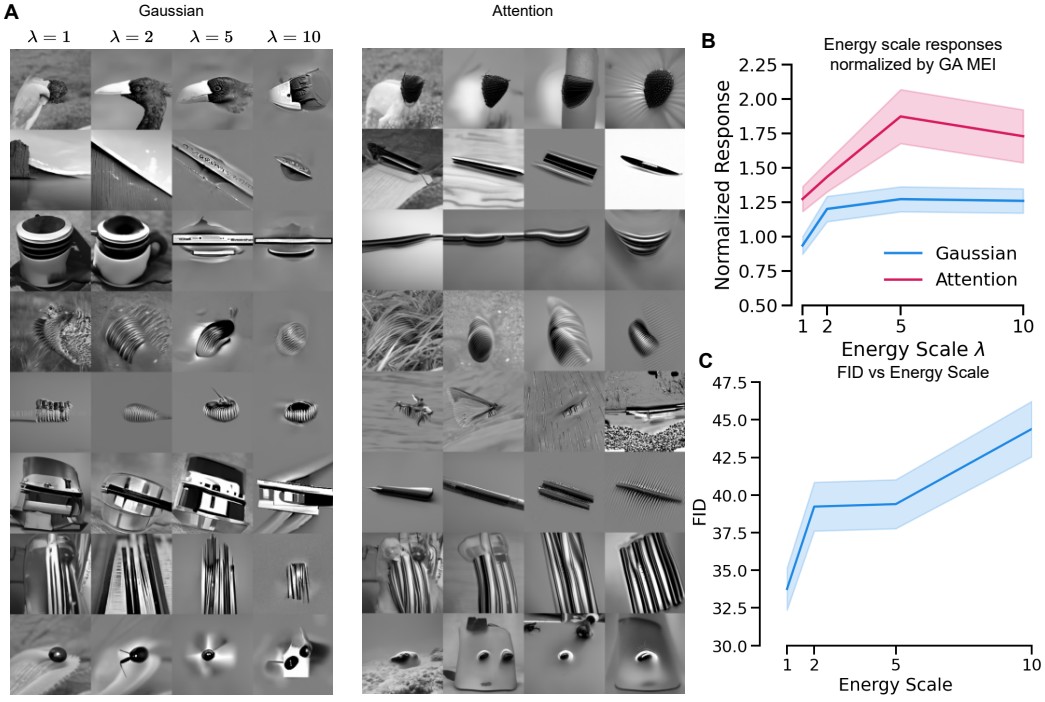

Figure 5: **a**) Examples of MENIs optimized using EGG diffusion in the macaque V4 for different neurons and different energy scales $\lambda \in \{1, 2, 5, 10\}$. **b**) Mean and standard error of the activations of neurons across different energy scales normalized by the activation of the GA MEI. **c**) FID between MEIs across energy scales and top-5 activating ImageNet images.

is a substantial gain, as Willeke et al. [24] required approximately 1.25 GPU years to optimize the MEIs presented in their study. With EGG, only approximately 0.25 GPU years would be needed to produce the results of the study, while providing higher quality and higher resolution MEIs. Thus, EGG can provide major savings in time and energy, *and* improve the quality of MEIs.

**Controlling the "naturalness" to generate most exciting natural images**   Unlike GA, EGG can also be used to synthesize more natural-looking stimuli by controlling the energy scale hyperparameter $\lambda$. Changing the value of $\lambda$ trades off the importance of the maximization property of the image and its "naturalness". To demonstrate this, we generated images for 150 neurons with the highest correlations to the average for the Gaussian model. We used energy scales $\lambda \in \{1, 2, 5, 10\}$, fixed the $100 \times 100$ image norm to 50, and used 50 steps re-spaced from 1000. Each image was generated using 3 different seeds and the best-performing image on the generator model was selected.

We show examples of the generated images across different energy scales in figure 5A for both the Gaussian model and the Attention model. For more examples, refer to the supplementary material (Fig. S6, Fig. S7). We subsequently quantified the predicted responses across different values of $\lambda$. We find that increasing $\lambda$ increases the predicted responses (Fig. 5B), however, at higher $\lambda$ values

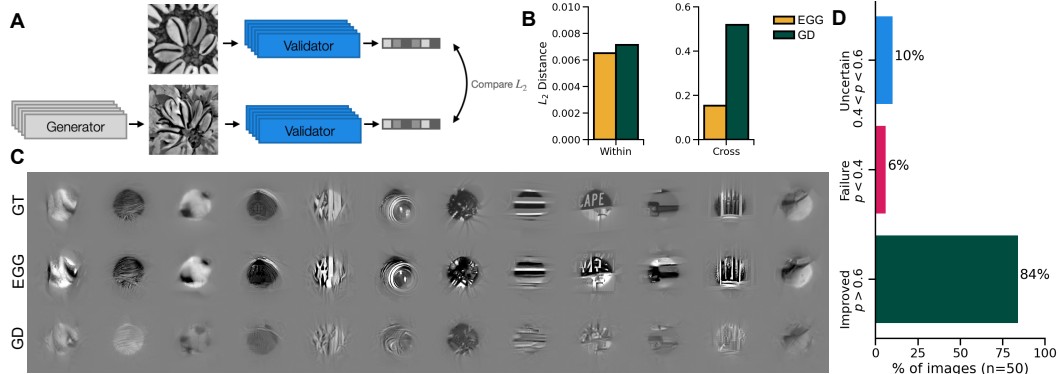

Figure 6: **a**) Schematic of the reconstruction paradigm. The generated image is compared to the ground truth image via $L_2$ distance in the unit activations space. Reconstructions from 1,244 units. **b**) $L_2$ distances in the unit activations space for the Within and Cross architecture domains comparing the EGG and GD generation methods. Shows that the EGG method generalizes better than GD across architectures. **c**) examples of reconstructions generated by EGG and GD in comparison to the ground truth (GT). **d**) Survey results on 45 voluntary human participants. Indicates that in 84% of images, the participants preferred the EGG generated reconstructions with a rate $\geq 0.6$.

the responses begin to plateau, or even decrease. Therefore, for generating MEIs, we use $\lambda = 10$ for the Gaussian model and $\lambda = 5$ for the Attention model. It can be further observed that decreasing $\lambda$ increases the naturalness of the generated image while preserving the features of the image that the neuron is tuned towards. To quantify the increase in the naturalness of the MEIs across $\lambda$s, we measured the FID score between the generated images at different $\lambda$ values and the top-5 ImageNet images (Fig. 5C). Our results show that by changing $\lambda$ we approach the natural images manifold (lower FID). We also find that EGG generates MEIs ($\lambda = 1$) similarly activating to the top-1 Imagenet images (Fig. S8).

**Image reconstruction from unit responses** Another application of EGG diffusion is image reconstruction from neuronal responses. A similar task has been attempted with success using diffusion models from human fMRI data [34, 35]. Given that only a small fraction of neurons were recorded, the image is encoded in an under-complete, significantly lower-dimensional space. Therefore, it is to be expected that the reconstructed image $x$ will not necessarily be equal to the ground truth image $x_{gt}$. However, a better reconstruction $x^*$ is one that generalizes across models. Therefore, regardless of the model $f$ used, we should get $||f(x^*) - f(x_{gt})||_2 = 0$. This is trivially true for $\bar{x}_0 = x_{gt}$ but, given the complexity of the model, there are likely other solutions. We therefore consider a masked version of the reconstructions for visualization. We mask the reconstructions to the joint receptive field of all 1,244 neurons. The mask is obtained by computing the average absolute gradients $\text{mask} = \mathbb{E}_x[|\nabla_x f(x)|]$ across the responses to the test images. The masks were normalized to be between 0 and 1 and the values below 0.25 are clamped to 0.

We can reconstruct images in the EGG framework by defining the energy function as an $L_2$ distance between the predicted responses to the ground truth image $f(x_{gt})$ and the predicted responses to a generated image (Fig. 6A) $E(x) = ||f(x) - f(x_{gt})||_2$. Note that, instead of $f(x_{gt})$, we could also use recorded neuronal responses. The images are generated from the Gaussian model with $\lambda = 2$ and 1000 timesteps, with the norm of the $100 \times 100$ image fixed to 60. We compare EGG to a gradient descent (GD) method that simply minimizes the L2 distance. The GD uses an AdamW optimizer with a learning rate of 0.05. In GD, at each optimization step the image $x_t$ is Gaussian blurred and the norm is set to 60 before passing it to the neural encoding model. We optimize the GD reconstruction up to the point where the train $L_2$ distance is matched between the GD and the EGG for a fair comparison of the generalization capabilities. We verified that the GD images do not improve qualitatively with more optimization steps (Fig. S9) We find that when generating the reconstruction using EGG diffusion we obtain 1) comparable within-architecture generalization and 2) much better cross-architecture generalization (Fig. 6B). The EGG-generated images produce lower within architecture distances for 84% of the images and for 98% in the cross-architecture case.

We show examples of EGG diffusion and GD reconstructions in Fig. 6C. Qualitatively, the images optimized by EGG resemble the ground truth image much more faithfully than the GD images. More examples are available in the supplementary materials (Fig. S10) and for reconstructions from the attention model see figure S11. We furthermore reconstruct images from real neurons by minimizing the distance between the model responses and recorded average neural responses to the image (Fig. S12).

**Human perceptual evaluation**    As shown in Cobos et al. [67], metrics like SSIM are not necessarily a good predictor of how well neuronal responses are reproduced in vivo. Therefore, we conducted a voluntary anonymous survey with 45 voluntary participants on 50 test images (Fig. S13). The participants were instructed to choose which image (GD optimized or EGG generated) was more similar to the ground truth image. Results show an 82.22% average preference for EGG-generated images (95% confidence interval [80.59%, 83.75%]; Wilson score interval). In 84% of the images, the EGG method was preferred more than 60% of the time (Fig. 6D).

**Limitations**    While EGG diffusion on average performs better than GA, it does come with limitations. Firstly, It is important to note that the results shown in this study have not been verified in vivo. Moreover, while the energy scale provides additional flexibility, it is important to keep it mind that it is an additional hyperparameter that needs to be selected to obtain the desired results, for which careful controls are necessary before the method can be used for in-vivo verification. For example, if the energy scale is too high, our method can be unreliable, which we identified in 3 out of 90 cases where EGG diffusion failed to provide a satisfactory result with the Gaussian model (Fig. 7). Furthermore, the parameter value also does not necessarily represent the same value across energy functions. Another limitation is that the maximal number of steps to generate the sample is constricted by the pre-trained diffusion model, i.e. at most 1000 steps can be used. In addition, the encoding model needs to generalize to the manifold of the diffusion model. Finally, since neurons are strongly driven by contrast, encoding models often tend to push generated images to very high contrast values. To avoid this effect and make the model focus on the image content, we evaluate the energy guiding encoding model at a normalized image to eliminate the contrast direction from the guidance. This can be seen as an additional prior.

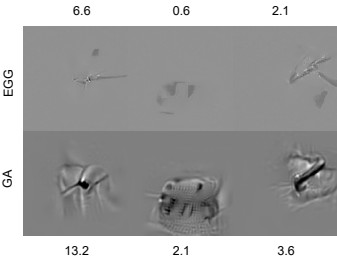

Figure 7: Examples of failure cases in comparison to the gradient ascent method. The text shows the predicted response rate by the within-architecture validation.

## 4    Discussion

In this study, we introduced a new model architecture, called the attention readout, for predicting the activity of neurons in macaque area V4 , which together with a fully data-driven convolutional core outperforms previous task-driven models. Furthermore, we propose a novel method for synthesizing images based on guiding diffusion models via energy functions (EGG). Our results indicate that EGG diffusion produces most exciting inputs (MEIs) which generalize better across architectures than the previous standard gradient ascent (GA) method. In addition, EGG diffusion significantly reduces compute time enabling larger-scale synthesis of visual stimuli. EGG diffusion is not limited to the generation of MEIs and, within the same framework, allows, among other characterizations, to 1) generate most exciting naturalistic images which approach the manifold of most activating images in the ImageNet database, and 2) reconstruct images from unit responses, which generalize better across architectures and qualitatively resemble more the original image than images obtained via regular gradient descent optimization. While the dataset we use for this study was recorded from the macaque visual cortex, it is in principle possible to use EGG for MEI generation and reconstructions with calcium imaging similar to the GA method on two-photon data in Walker et al. [17]. In fact, EGG can be applied to any modality that yields an encoding model and where a suitable diffusion model is available. More generally, EGG can be used whenever the "constraint" on a particular image can be phrased in terms of an energy function. In summary, EGG diffusion provides a flexible and powerful framework for studying coding properties of the visual system.

## Acknowledgments

The authors thank the International Max Planck Research School for Intelligent Systems (IMPRS-IS) for supporting Konstantin Willeke and Arne Nix. The authors also thank Mohammad Bashiri and Suhas Shirinvasan for their technical support and helpful discussions. The research was supported by the Cyber Valley Research Fund (AN, FHS). FHS is further supported by the German Federal Ministry of Education and Research (BMBF) via the Collaborative Research in Computational Neuroscience (CRCNS) (FKZ 01GQ2107), as well as the Collaborative Research Center (SFB 1233, Robust Vision). PP is supported by the German Federal Ministry for Economic Affairs and Climate Action (FKZ ZF4076506AW9). We also acknowledge support from the National Institute of Mental Health and National Institute of Neurological Disorders And Stroke under Award Number U19MH114830 and National Eye Institute award numbers R01 EY026927 and Core Grant for Vision Research T32-EY-002520-37 as well as the National Science Foundation Collaborative Research in Computational Neuroscience, USA with grant number IIS-2113173, Germany with FKZ: 01GQ2107. We thank our lab members, family, and friends for participating in the anonymous survey.

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

# A  Supplementary Material

## A.1  Training Data

Electrophysiological data were acquired as broadband signal (0.5Hz-16kHz), from a pair of male rhesus macaque monkeys (*Macaca mulatta*), using 32 channel linear silicon probes (NeuroNexus V1x32-Edge-10mm-60-177). The data was spike-sorted, and single units were isolated based on unit stability, refractory periods, and channel principal component pairs. Visual stimuli were presented to the animals on a 16:9 widescreen HD LCD monitor at 100c̃m viewing distance. The animals were rewarded with juice if they maintained their gaze around a red fixation target throughout each trial. At the beginning of each recording session, the receptive fields (RFs) of the neurons were mapped in relation to a fixation target using sparse random dot stimuli, and the population RF was pulled towards the center of the screen by adjusting the fixation target. A collection of 24,075 images from ImageNet [68] was transformed into gray-scale and cropped to the central $420^2$ px and had 8 bit intensity resolution. These images were presented as visual stimuli during standalone generation recordings of 1244 units and during closed-loop recordings of 82 units. For details on the closed loop paradigm, please refer to Willeke et al. [24].

## A.2  Supplementary Experiments

### A.2.1  Attention Model

**Attention readout uses its ability to shift receptive fields**   Receptive fields of neurons in area V4 can shift before the onset of saccades, believed to be associated with attentional shifts [26]. We investigate whether the Attention model actually uses its ability to shift the receptive field depending on image content. We inspect the attention mask of the attention model. We compute the center of mass of the upper 5% percentile of each attention mask. We then compute the average distance between the center of masses across different images for each neuron. We plot the average distance against the test correlation of each neuron observing that the attention readout does perform shifts (Fig. S1a). We also show qualitative examples of the masks and the means in Fig. S1b.

**Centered Kernel Alignment**   We computed CKA of the neural encodings across architectures between the Attention model and Gaussian model and within architecture between different seeds (e.g. Attention 1 and Attention 2 are models with the same architecture, but trained with different seeds). The CKA is computed between the predicted neuronal responses. We observe that the within-architecture similarity is very high (> 0.99) for both architectures and the cross architecture similarity is slightly lower, but also high (> 0.9) (Table 2). We expect such an outcome, since both architectures were trained to model the same neural representation.

| Model | Attention 1 | Attention 2 | Gaussian 1 | Gaussian 2 |
|---|---|---|---|---|
| Attention 1 | 1 | 0.9949 | 0.9133 | 0.9116 |
| Attention 2 | 0.9949 | 1 | 0.9145 | 0.9129 |
| Gaussian 1 | 0.9133 | 0.9145 | 1 | 0.9994 |
| Gaussian 2 | 0.9116 | 0.9129 | 0.9994 | 1 |

Supplementary Table S1: Centered Kernel Alignment for the two architectures comparing across and within architectures.

### A.2.2  MEI generation

**Comparison of experimental setups**   For the GA optimization, we use the established method for generating MEIs that has been tested in vivo Walker et al. [17]. However, we perform a comparison study to show that the parameters chosen are selected to maximize the performance of the GA method. We rerun the MEI optimizations using the AdamW optimizer and find a significant decrease in performance in comparison to the SGD optimizer (r = 0.69). We also run the MEIs for 100 steps instead of 1000 and also find a performance decrease (r = 0.95) (Fig. S2).

**Generating MEIs in color**   The diffusion model generates color images, so in principle, it can generate color MEIs. We attach some examples (Fig. S4). Since the encoding models are trained

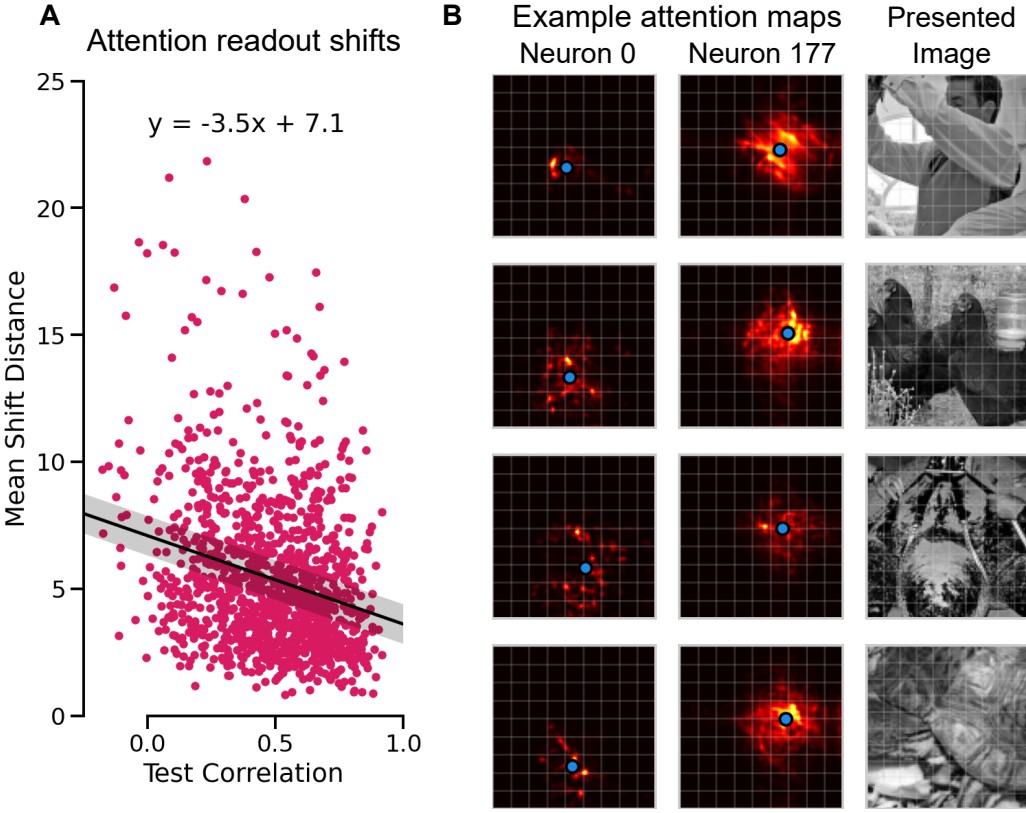

Supplementary Figure S1: **a**) Comparison of the mean shift distance between the center of mass for the attention masks against the test correlation of the neurons. **b**) Example attention maps of neurons responding to different neurons. The blue dot shows the center of mass. The examples show that the neuron shifts its center of mass.

on grayscale images because the animals only saw grayscale images the colors in these may not be meaningful. However, if one were to use color stimuli it would be possible to generate MEIs that are colored and potentially meaningful.

**MENIs vs ImageNet search** We compare the generated MEIs ($\lambda = 1$) to a standard approach for finding natural images for individual neurons. To that end, we perform a search across 100k images from the ImageNet dataset [68] to find the top-1 most activating image for a particular unit. We then compare the predicted activations of the top-1 ImageNet image and the generated MEIs ($\lambda = 1$) in the cross-architectures paradigm (Fig. S8). We find that the generated MENIs drive comparable activation to the top-1 ImageNet images. Like in the MEI generation paradigm, EGG can thus significantly speed up the search for activating natural images, as it does not need to search through millions of images.

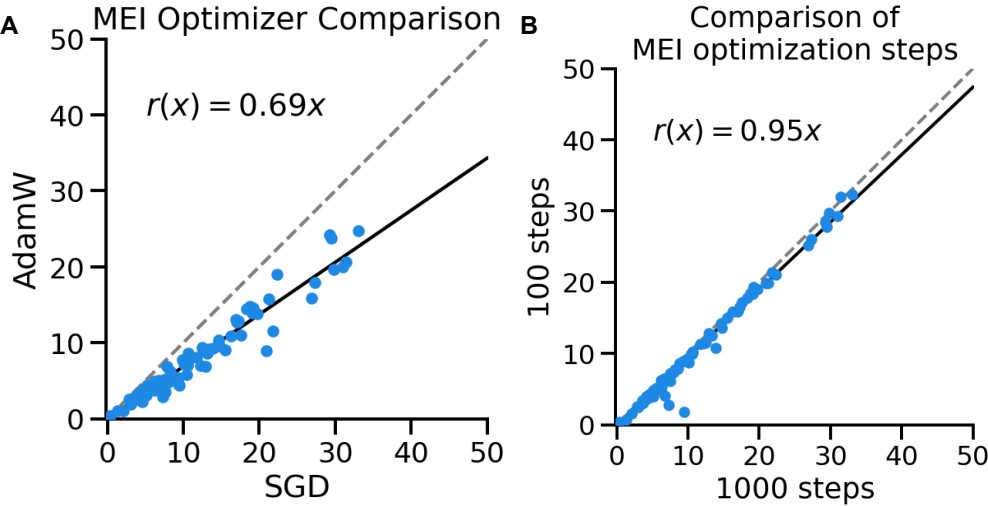

Supplementary Figure S2: Comparison of different experimental setups of MEI optimization using the GA method. **a**) Use of SGD vs AdamW optimizer. The SGD optimizer outperforms the AdamW optimizer on within architecture evaluation. **b**) Increasing the number of steps slightly decreases the within architecture performance.

**Gaussian**     **Attention**

Seed 0     Seed 1     Seed 2

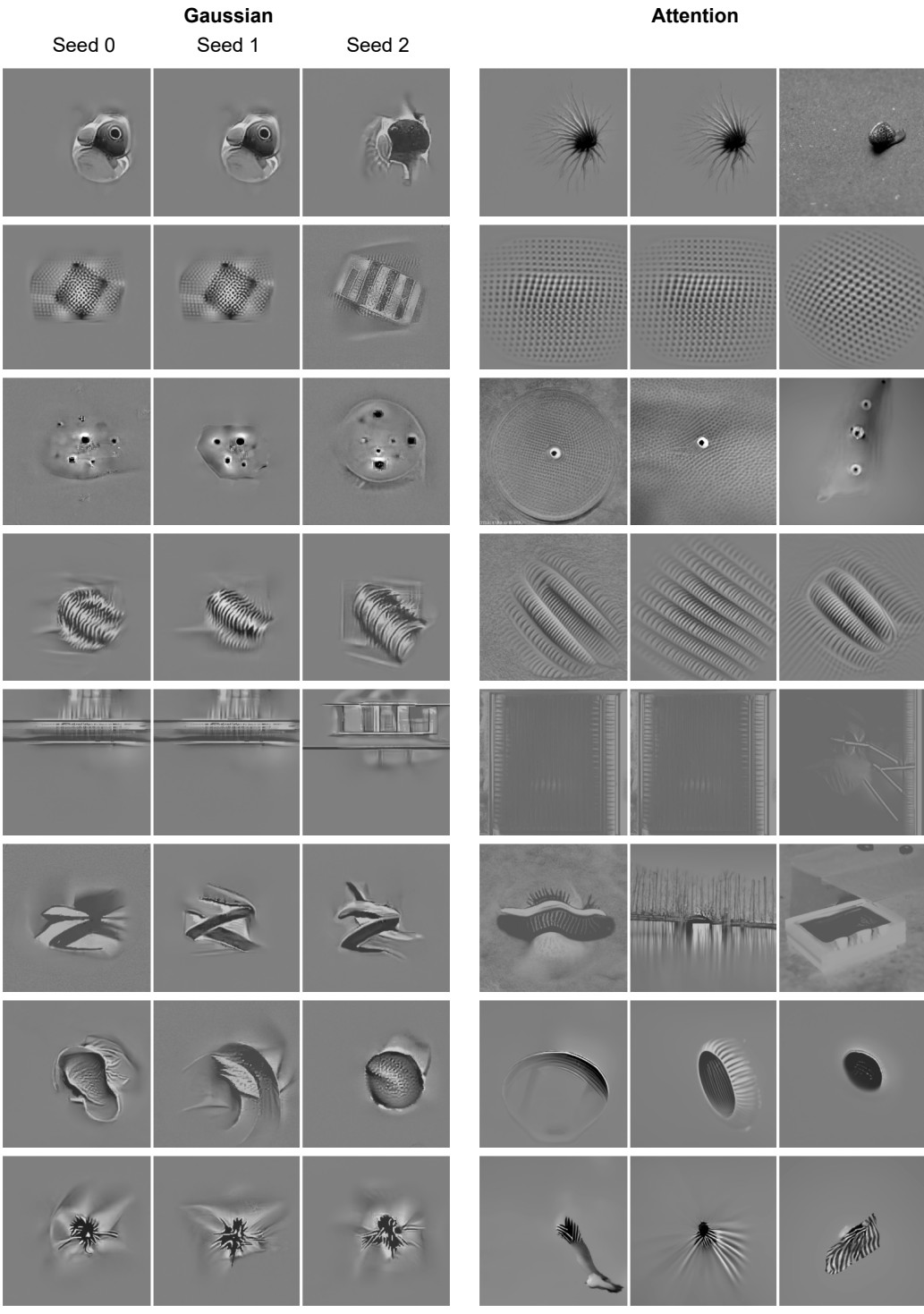

Supplementary Figure S3: Variability dependent on the seed used for generating MEIs in the Gaussian model and the Attention model. Each column represents a different seed and each row a different neuron. Results shown for the Gaussian and Attention models.

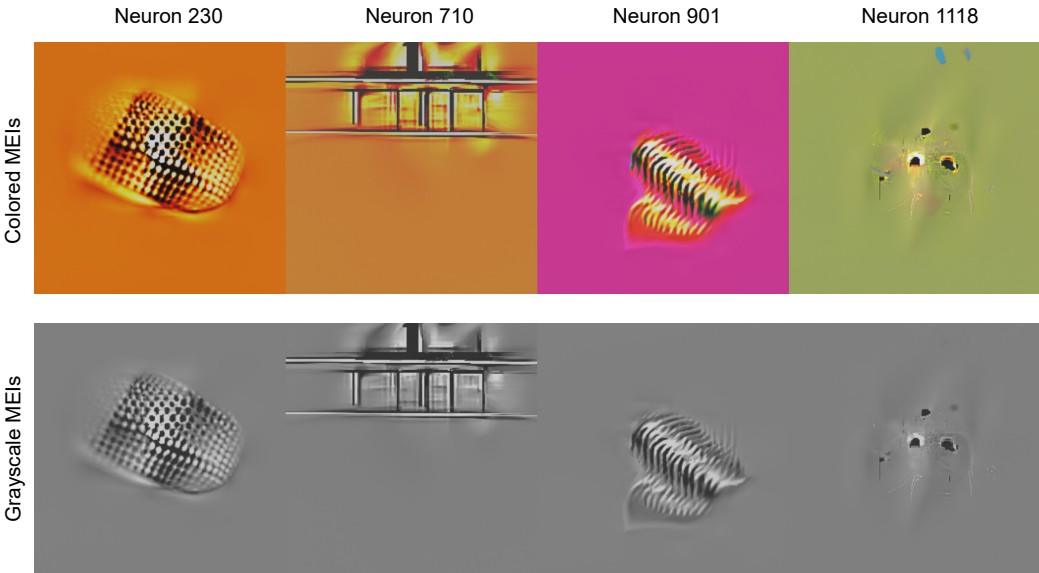

Supplementary Figure S4: Examples of MEIs in their original color version (before converting to grayscale). Top row is the direct output RGB images from the diffusion model, the bottom shows the grayscaled version. Each column corresponds to a different neuron.

Gaussian

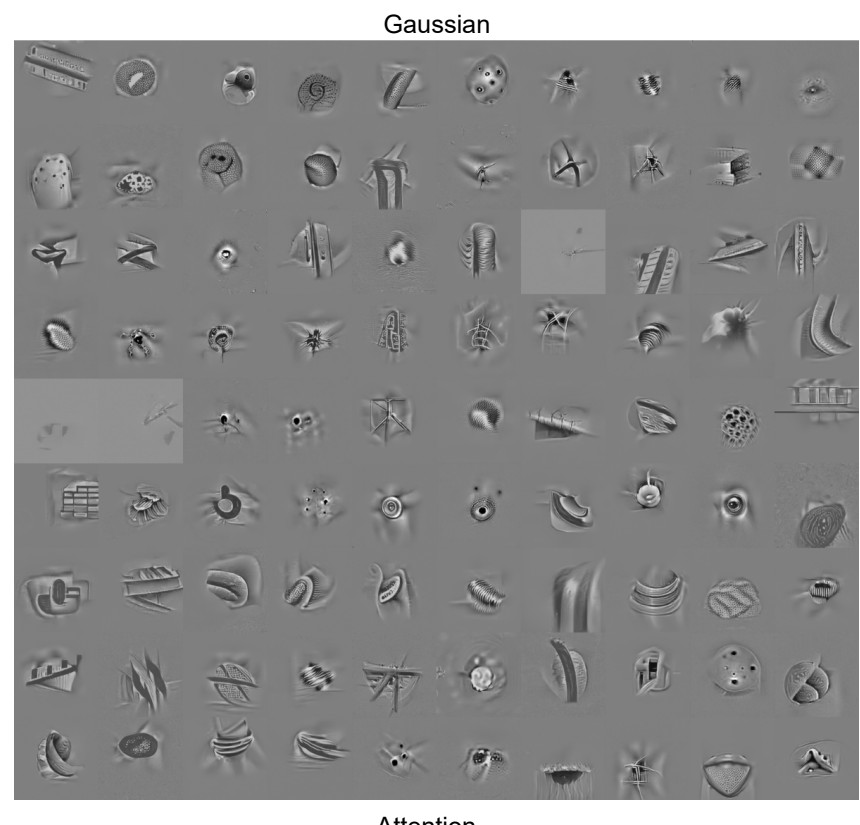

Attention

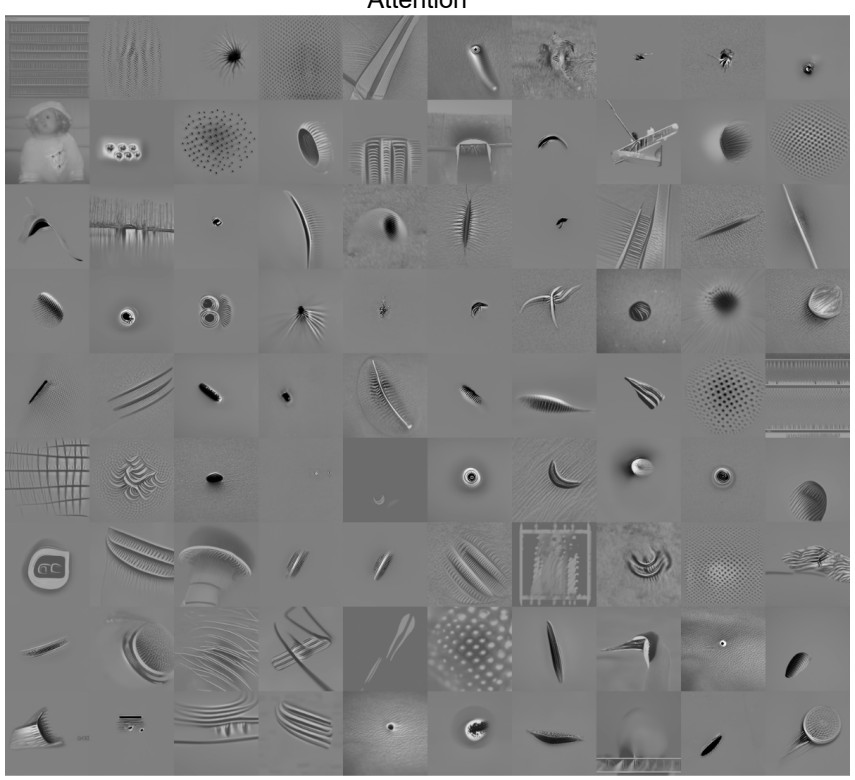

Supplementary Figure S5: Examples of MEIs generated using EGG for the Attention and Gaussian models.

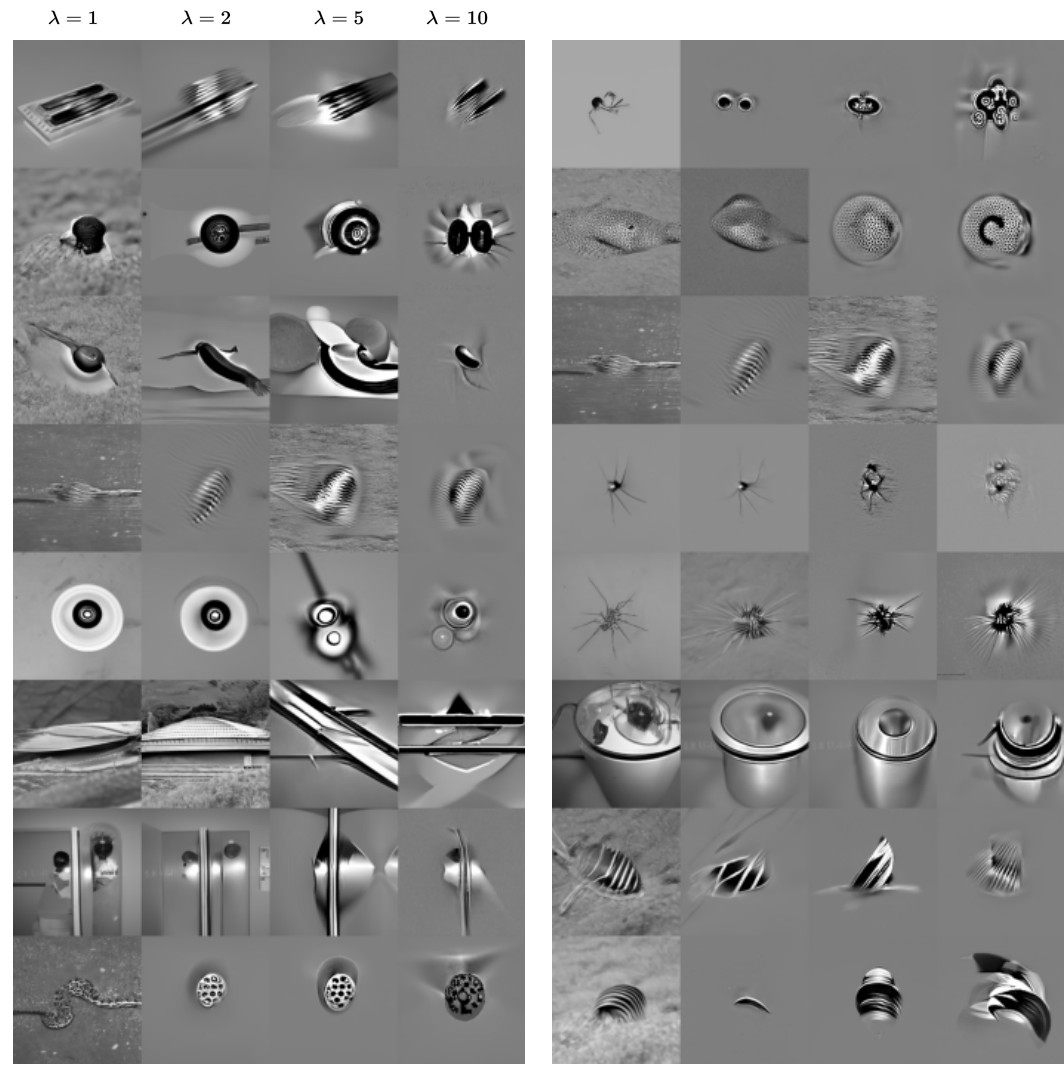

Supplementary Figure S6: Examples of images generated using EGG diffusion in the Monkey V4 with different energy scales $\lambda \in \{1, 2, 5, 10\}$. Generated for the Gaussian model. Units not matched with the images shown for the Attention model.

$\lambda = 1$ $\qquad$ $\lambda = 2$ $\qquad$ $\lambda = 5$ $\qquad$ $\lambda = 10$

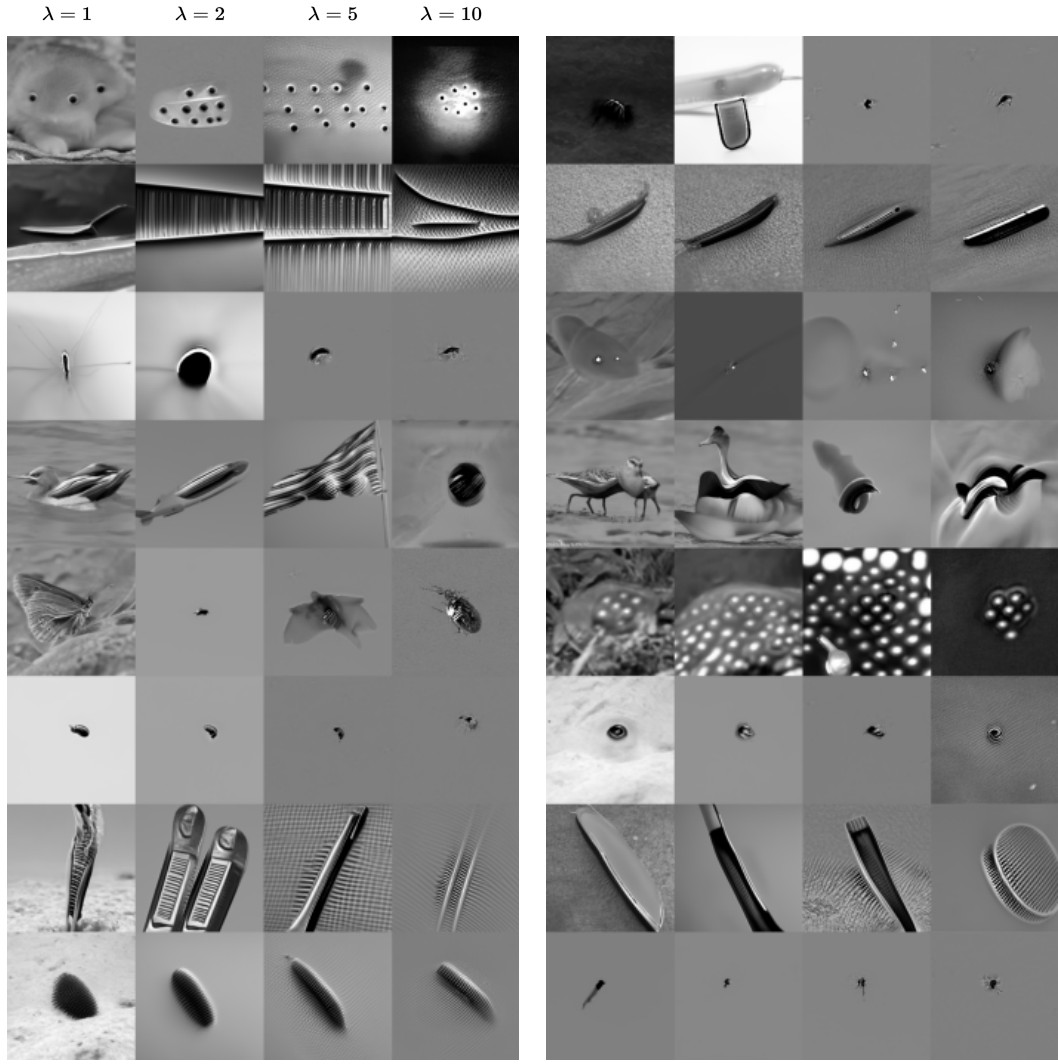

Supplementary Figure S7: Examples of images generated using EGG diffusion in the Monkey V4 with different energy scales $\lambda \in \{1, 2, 5, 10\}$. Generated for the Attention model. Units not matched with the images shown for the Gaussian model.

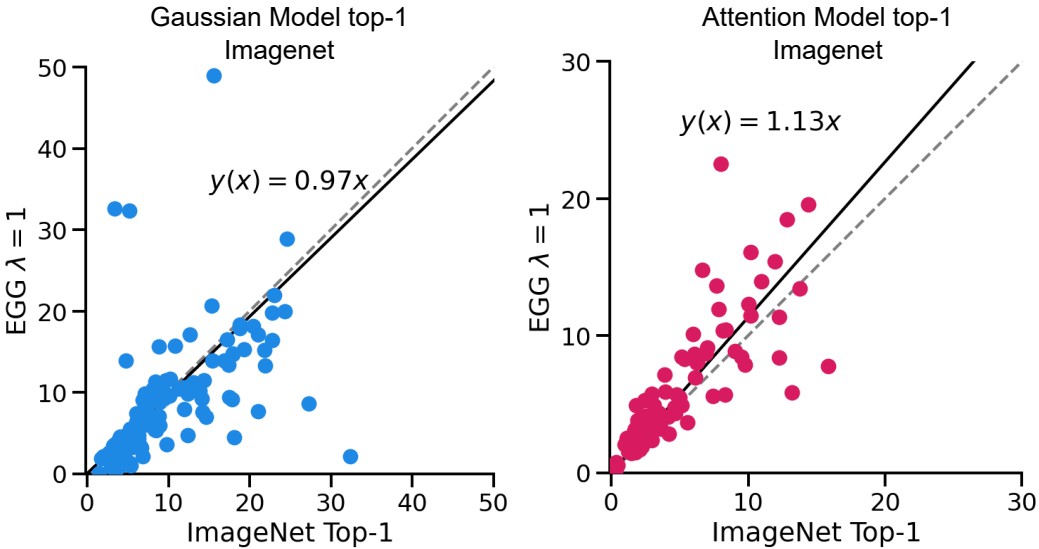

Supplementary Figure S8: Comparison of the MEIs $\lambda = 1$ activations to the top-1 most activating ImageNet images per neuron in the cross-architecture domain. Line fit obtained via Huber regression with $\varepsilon = 1.1$. In the left panel, three points at $(11, 65)$, $(9, 70)$, and $(16, 120)$ are not shown for visualization purposes.

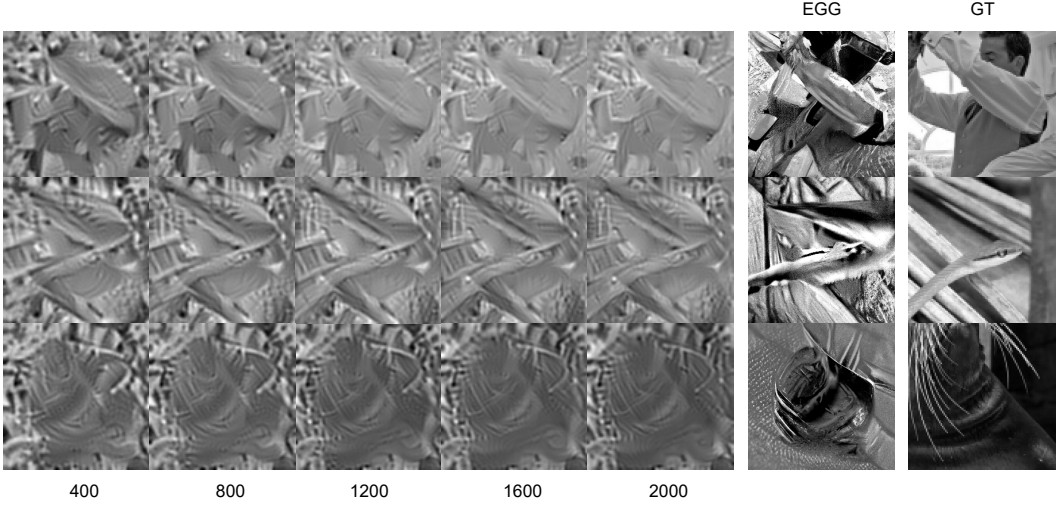

Supplementary Figure S9: Examples of reconstructions using GD across various training lengths. Increasing the training does not bring the generated image closer visually to the GT nor EGG.

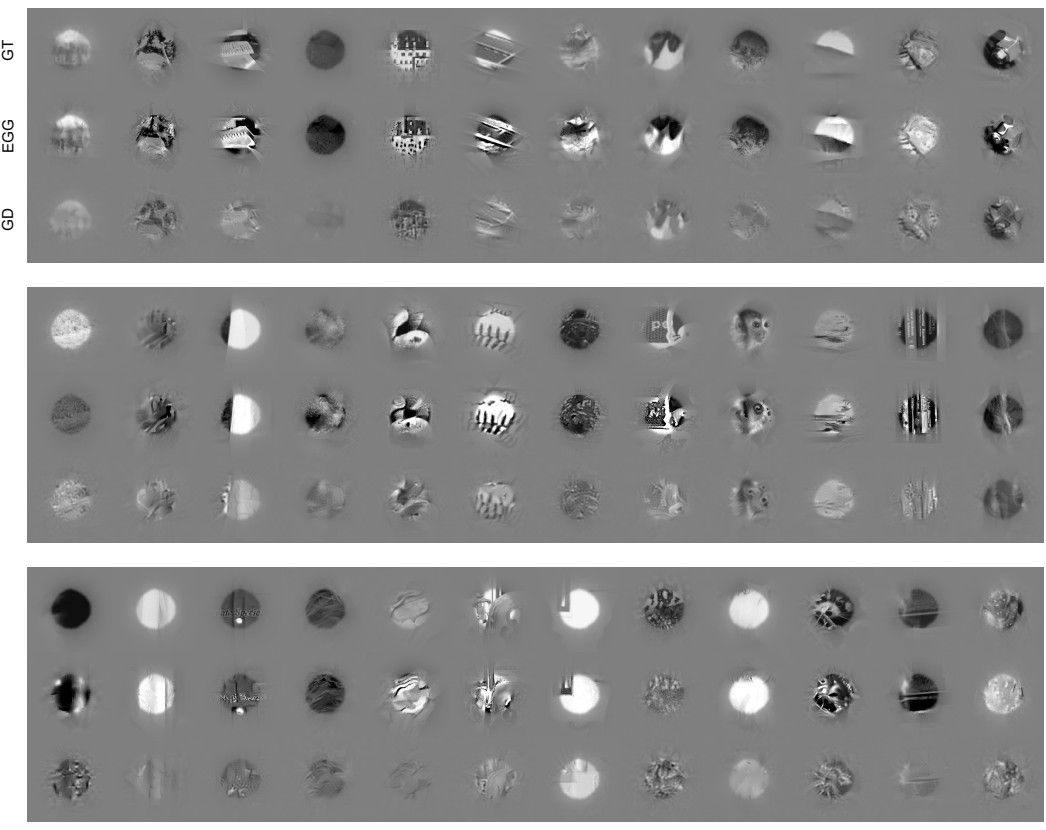

Supplementary Figure S10: Reconstruction examples from the Gaussian model. Generated using EGG diffusion and gradient descent.

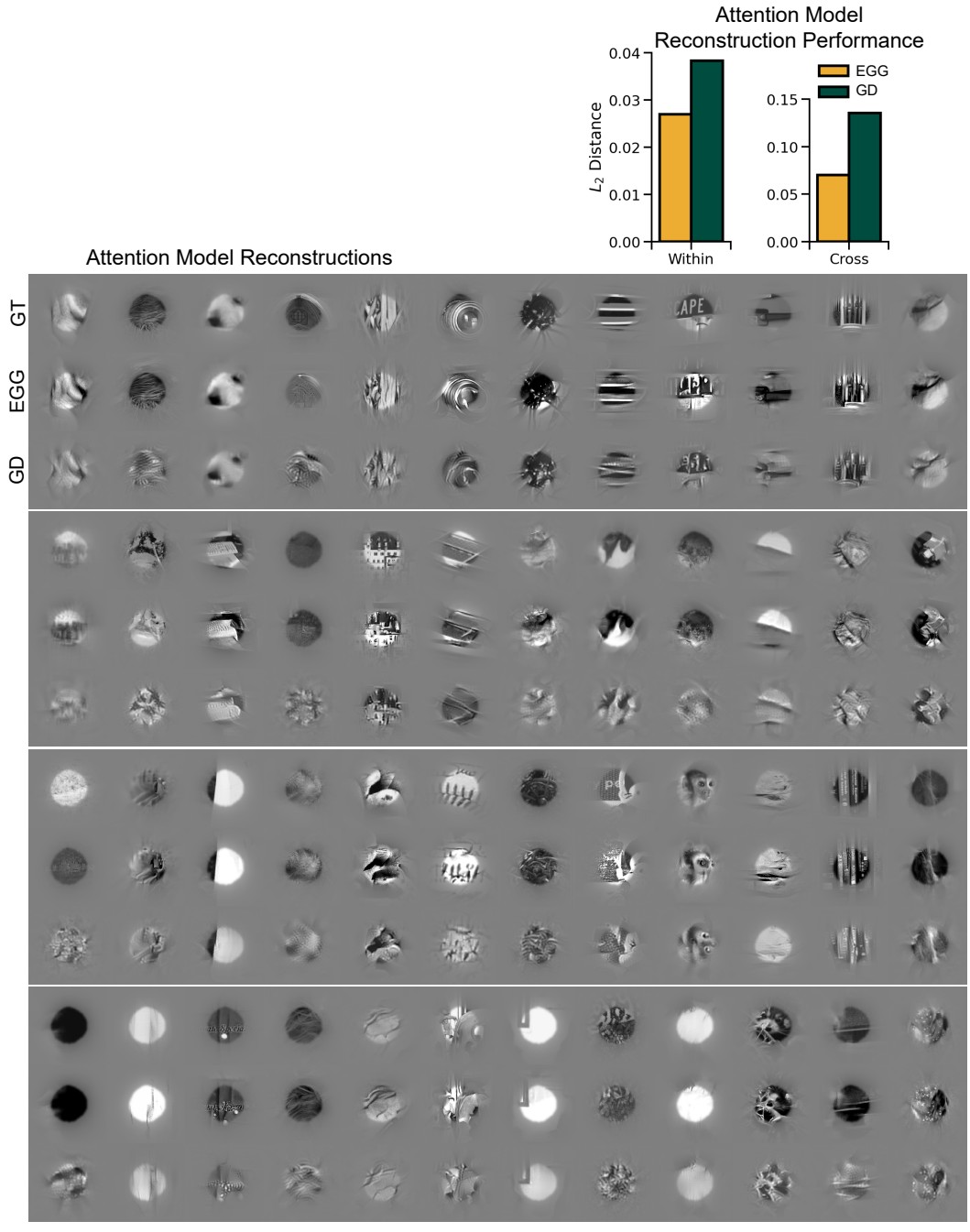

Supplementary Figure S11: Reconstructions from the Attention model. Top row in each panel is the ground truth image, middle is our EGG generated reconstruction and last row is the GD optimized reconstruction. Bar plot shows the performance of both EGG and GD in the within and cross architecture paradigms in terms of $L_2$ distance.

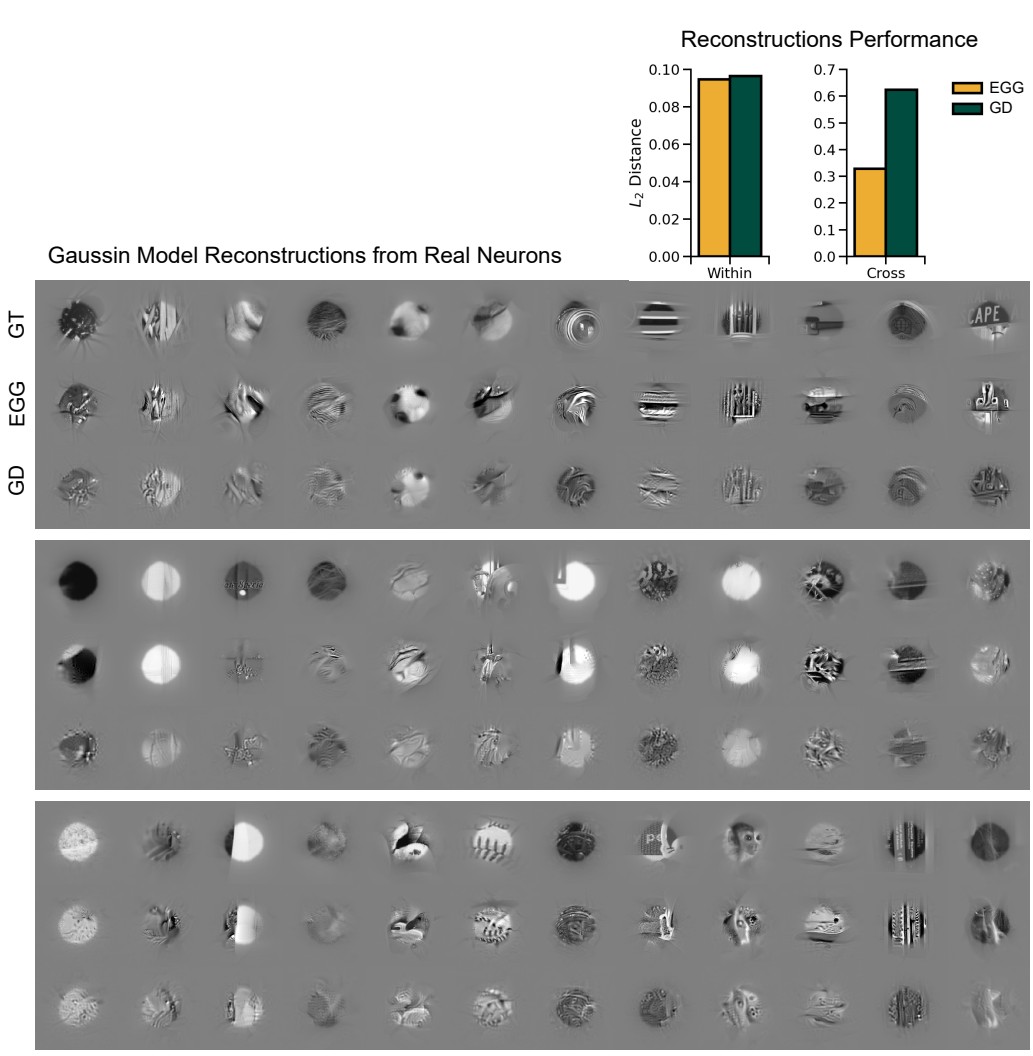

Supplementary Figure S12: Reconstructions from real neurons using the Gaussian model. Top row in each panel is the ground truth image, middle is our EGG generated reconstruction and last row is the GD optimized reconstruction. Bar plot shows the performance of both EGG and GD in the within and cross architecture paradigms in terms of $L_2$ distance.

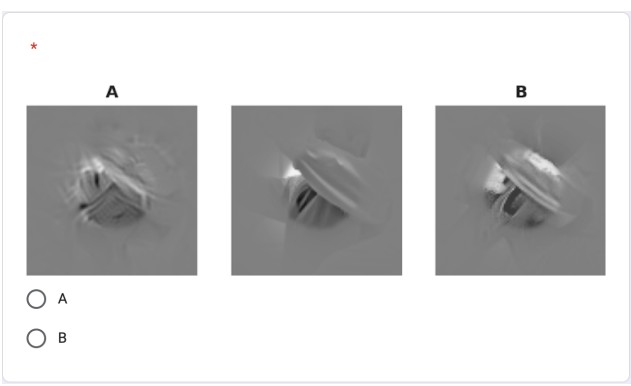

Supplementary Figure S13: Setup for the human perceptual evaluation. The voluntary participants were presented with 50 images with the GT image always in the middle and the GD or EGG reconstructions were placed randomly on each side of the GT image. The participants were provided with the question "Which of the images looks more like the image in the center?". They were provided with context text: "In our study, we are reconstructing images from the brain activity. We have two methods to do so and we want to find out which one looks better to the human eye. Your participation is entirely voluntary, and you have the right to withdraw at any time without providing a reason. Please note that all responses will be kept confidential, and the data collected will be used solely for research purposes. Your identity will remain anonymous, and your personal information will not be disclosed to anyone."

