# OpenReview forum: "Energy Guided Diffusion for Generating Neurally Exciting Images"
_NeurIPS.cc/2023/Conference — NeurIPS 2023 poster_

### Official Review · Reviewer_BnXF · 2023-06-23

**Soundness:** 3 good
**Presentation:** 3 good
**Contribution:** 2 fair
**Rating:** 6
**Confidence:** 3

**Summary:**

In this work, the authors first employed attention readout to train a model for predicting neural responses ($y$) from image ($x$), aiming to address the issue of attention effects in the V4 area. Subsequently, they applied the image ($x$) from text ($y$) method proposed by Prafulla [1] to generate images ($x$) from neural responses ($y$).

Reference:

[1] Prafulla Dhariwal and Alex Nichol. Diffusion models beat GANs on image synthesis. May 2021.

**Strengths:**

Compared to previous research, this study incorporates two technologies to enhance the performance of their model in predicting neuronal responses and MEIs. The first technology involves integrating an attention map into their encoding network. From a biological standpoint, the attention method allows for the receptive field of neurons to be dynamically adjusted based on different inputs, which aligns with observations made in experiments on V4. Empirically, attention-based Vision Transformer (ViT) models have shown superior performance compared to ResNet models in the field of computer vision. In this study, the performance of the proposed model surpasses the current state-of-the-art in predicting neuronal responses.

The second technology employed in this research is the use of Bayesian diffusion methods to decode input from neural responses. The diffusion model is renowned for its ability to generate intricate image details, and the authors demonstrate that their approach, called EGG, outperforms the previous GA method.

Although the authors did not develop these two methods themselves, it is possible that their application in neural data analysis is novel. To the best of my knowledge, this study appears to be the first instance of using the attention map in the encoding network for predicting neuronal responses, taking into consideration the observed phenomenon in V4. Furthermore, the utilization of the Bayesian diffusion method, specifically the EGG approach, for decoding neural responses seems to be a novel application in this context.

**Weaknesses:**

1.The current images are grayscale. Can the MEI images be colored?

2. The current neural data are collected by silicon probes. Can we use techniques such as two-photon or single-photon imaging to record more neuronal data?

**Questions:**

How many neurons are used in EGG? 1,244 individual neurons?

---

> ### Author Rebuttal · Authors · 2023-08-08
>
> We appreciate your review. We respond to your concerns below. In case you have any more questions we would be happy to discuss them.
>
> ### RE 1: Color MEIs
> The diffusion model generates color images, so in principle, it can generate color MEIs. We attach some examples (Fig. B). Since the encoding models are trained on grayscale images because the animals only saw grayscale images the colors in these may not be meaningful. However, if one were to use color stimuli it would be possible to generate MEIs that are colored and potentially meaningful.
>
> ### RE 2: Applicability to other Neural Experimental Techniques
> See general response **RE 3:  Applicability to other Neural Experimental Techniques**
>
> ### RE Q1: How many neurons are used in EGG?
> Yes, we use 1,244 individual neurons. For each MEI a single neuron is selected and for the reconstructions, we use all 1,244 neurons.

---

### Official Review · Reviewer_H4Az · 2023-06-28

**Soundness:** 2 fair
**Presentation:** 3 good
**Contribution:** 2 fair
**Rating:** 5
**Confidence:** 3

**Summary:**

The authors tackle the problem of synthesizing most exciting inputs for neurons in the higher visual cortex (V4 in their case) of macaque, with data collected using electrophysiology.

The paper makes two claimed contributions:

1. The authors propose a new encoding architecture which uses a data-driven CNN core with a cross-attention read out layer. The cross-attention layer is parameterized similar to traditional cross-attention in other machine learning papers. In the author's design, there is a learned per-neuron query vector, and spatial key/value embeddings derived from pixel-wise linear projections of the CNN feature map that is shared for all neurons. The authors compare this against a task-optimized backbone with learned gaussian readout. The authors show that the the attention encoder performs better with high probability (via a Wilcoxon test) on novel non-training images.

2. The authors propose energy guided diffusion. Where by they modify the score-prediction with the derivative of vector valued function (not a score function corresponding to a well posed distribution). They propose a modification which does not require the energy function to take as input noisy images, they accomplish this by using the "pred_xstart" code provided by [1].

The authors validate their method by first comparing their method against gradient ascent + gradient blurring in MEI synthesis. They find that EGG is able to generate MEI in a faster fashion and with better cross-architecture generalization properties than GA.

In the MENI experiment, they show that a tradeoff between naturalness and excitation by adjusting the strength of the gradient of the energy function. They find that their synthesized MENI ($\lambda=1$) are roughly comparable to imagenet top-1 images in cross-arch predicted activations.

In the third experiment, they authors experiment with stimulus reconstruction. They accomplish this by modifying the energy function to minimize the L2 distance between predicted and ground truth neuron activities. They find that EGG regularized stimulus reconstruction is more faithful.

[1] Dhariwal, Prafulla, and Alexander Nichol. "Diffusion models beat gans on image synthesis." Advances in Neural Information Processing Systems 34 (2021).

In the supplementary, the authors provide additional descriptions of data collection (32 channel) and experiment design. The authors provide further experiments that compare ResNet & Attention encoders, the strength of $\lambda$, and how pure gradient methods compare with EGG in stimulus reconstruction.

**Strengths:**

The paper on balance is well written, the authors are largely clear in their experimental design and their evaluation. The authors provide sufficient detail in the paper itself for reproduction, with additional code provided in the supplemental. The code is well written and easy to follow.

The use of diffusion models to regularize the synthesis of most exciting inputs for monkey V4 collected using electrophysiology is novel, and to the author's knowledge it has not been attempted before.

The authors perform a variety of experiments, and I find their proposed design for the attention encoder and evaluation of the attention encoder to be convincing.

**Weaknesses:**

The authors did an good job writing a paper that has clarity and do a great job in providing details. But in my opinion, the authors overstate their contribution with regard to "Energy Guidance" (EGG). If the author can include additional citations, reduce their overly broad claims, and provide additional experiments/metrics, the paper would be improved significantly.


**The paper could stand out based on the experiments alone, but the authors have emphasized energy guidance to be a central contribution** without citing the vast number of papers in computer vision that have been published in the past two years that:
1. Similarly do not use a well-posed score function in the form of the derivative of a classifier. **This aspect is not novel.** In fact I would argue that most of gradient conditioning papers of diffusion models published today explicitly do not use a score function in the form of a classifier gradient. These papers are not cited.
2. Estimate a clean sample ($x_0$) and do not feed in a noise corrupted image to the model providing guidance. The paragraph in lines 162-174 seem to indicate that this approach is novel in the context of classifier/gradient guidance for diffusion models. However **similar approaches have been widely used in computer vision literature**. These papers are not cited.

* On contributions and prior work
  * GLIDE from 2021 [1] proposed to perform image synthesis again using the gradient of the dot product of a CLIP image vector and a text vector to modify the diffusion output. Note that GLIDE used a noisy trained version of CLIP to perform guidance, however in this approach it seeks maximization of a dot product output which does not yield a "proper" distribution score function when you take the gradient.

  * GLIDE and the DALL-E 2 [2] paper cite crowsonkb's 2021 open source CLIP guidance work [3, 4]. These two codebases combine CLIP guidance with the pred_x0 trick (eq 6 of this paper) without retraining the diffusion or gradient model. Similarly, the hugginface diffusers library minimize the orthodromic distance (derivative is not a proper score function) in CLIP space rather than maximizing a dot product using CLIP, and also use the pred_x0 trick without retraining the diffusion or gradient model. This estimated $x_0$ trick has also been formally described in [5] eq's 3 and 4. I suggest the authors cite at least one of these papers and clarify their contributions.

* On the soundness of the experiments

  * I also found some of the experimental setups to be inconsistent.
  * in the MEI experiment line 221, the naïve SGD optimizer is used, and this forms the basis of the claim in Figure 4 to show GA is much slower than EGG. However in the image reconstruction experiment line 282, the more sophisticated AdamW optimizer is used. There is no reason why the AdamW optimizer cannot be used with gaussian blur gradient conditioning via filtering at an higher stage of the backpropagation.
  * for the MEI experiment, GA is run for 1,000 steps, while EGG was run for 100 steps. This does not seem to be an entirely fair comparison time wise.
  * I'm not sure why the authors decided to normalize the image itself to 25 (line 230) for MEI, 50 (line 251) for MENI, and 60 (line 281) for reconstruction. This step seems to quite explicitly break the energy guidance output. Can you provide a justification in the text for why this is done, and why you use different norms for different tasks? Can the authors perform MEI/SGD/AdamW experiments without this step?

* On the lack of evaluation

  * There is a lack of quantitive evaluation metrics aside from predicted neural activity. The paper almost entirely relies on qualitative claims when it comes to MEI/MENI/Reconstruction output. There are no actual metrics to indicate that the MEI/MENI/Reconstruction outputs are more similar to the ground truth most exciting natural stimuli.
  * I suggest that the authors use common vision metrics like SSIM/PSNR/MSE to evaluate the low level similarity of the images, or high-level image metrics like perceptual loss (VGG), CLIP cosine distance, or distribution-wise comparisons like FID/CLIP-FID (see question 3 in section below). I don't think all the suggested metrics here are needed, but at least a few (any of SSIM/Perceptual VGG/CLIP for image reconstruction, and any of FID/CLIP-FID for MENI; or if the previous metrics are not possible, perhaps a human survey on Mechanical Turk/Prolific to evaluate if the images are better?) should be added where appropriate.

Overall I think the authors have presented an interesting system, but there is no citation or acknowledgement of prior work from computer vision which use non-classifier based gradient guidance of diffusion models, or using estimated $x_0$ to alleviate the need for noisy trained classifiers. Otherwise I think the paper is interesting and would be improved if the authors can clarify the scope of their contributions and better quantify their claims. I would happily re-evaluate if the authors can improve this paper in a subsequent revision.

[1] Nichol, Alex, et al. "Glide: Towards photorealistic image generation and editing with text-guided diffusion models." arXiv preprint arXiv:2112.10741 (2021).

[2] Ramesh, Aditya, et al. "Hierarchical text-conditional image generation with clip latents." arXiv preprint arXiv:2204.06125 (2022).

[3] https://github.com/afiaka87/clip-guided-diffusion/blob/c1d5906225586bc8455bb17c29a3c2caf9a02766/cgd/cgd.py#L141

[4] https://colab.research.google.com/drive/12a_Wrfi2_gwwAuN3VvMTwVMz9TfqctNj#scrollTo=X5gODNAMEUCR&line=41&uniqifier=1

[5] Li, Wei, et al. "UPainting: Unified Text-to-Image Diffusion Generation with Cross-modal Guidance." arXiv preprint arXiv:2210.16031 (2022).

**Questions:**

1. Could you clarify why the MEI experiment uses SGD, but the image reconstruction experiment uses AdamW?
2. For Figure 3, you make qualitative claims that the EGG MEIs are better, can you back this up with quantitive numbers like SSIM/Perceptual loss (VGG)/Inception/CLIP distance against the natural input that most excites the neuron?
3. For Figure 5, can you characterize the distributional similarity of the images using standard image metrics like Fréchet inception distance or CLIP-FID proposed by Kynkäänniemi (MENI vs top-k of natural images for a neuron)? Something like the Figure 4 of Imagen [1] Figure 4's pareto curves which measure how the energy scale affects the image distribution distance (FID/CLIP). For Q3/Q4, if the image metrics are not possible, perhaps a human survey on Mechanical Turk/Prolific to evaluate if the images for the experiments are more similar/better.
4. For Table 1, Figure 3B, Figure 5C, and Figure 6B, could you clarify what is the "base" model, as in which model you use for image synthesis, and which model is the evaluating model?
5. Can you clarify the solver you use in the diffusion model? From the code, it seems like you use the DDPM solver, however there are a variety of stronger solvers (DDIM/PNDM/DPM-solver [2,3,4]) which yield convergence in as few as 10 steps. Is there any reason you decided to go with such an old solver?

Overall I think the clarity of the paper is good, but could be further improved with a few small clarifications and incorporation of standard vision metrics (SSIM/VGG perceptual loss/FID/CLIP-FID).

[1] Saharia, Chitwan, et al. "Photorealistic text-to-image diffusion models with deep language understanding." Advances in Neural Information Processing Systems 35 (2022): 36479-36494.

[2] Song, Jiaming, Chenlin Meng, and Stefano Ermon. "Denoising diffusion implicit models." arXiv preprint arXiv:2010.02502 (2020).

[3] Liu, Luping, et al. "Pseudo numerical methods for diffusion models on manifolds." arXiv preprint arXiv:2202.09778 (2022).

[4] Lu, Cheng, et al. "Dpm-solver: A fast ode solver for diffusion probabilistic model sampling in around 10 steps." arXiv preprint arXiv:2206.00927 (2022).

**Limitations:**

The authors clearly describe the limitations of their experiments.

---

> ### Author Rebuttal · Authors · 2023-08-08
>
> We appreciate your helpful review. Please find our responses to your questions below. If you have any further questions we are happy to discuss.
>
> ### RE: Prior work and scope
> We will include and discuss the additional prior work, and make sure to make it even clearer that we do not claim to have invented gradient conditioning and the clean sample trick.
>
> GLIDE and DALLE-2 use a different trick, their model directly predicts x_0, we predict $\epsilon$ and get an approximate x_0 from x_t and $\epsilon$. Crowsonkb's work indeed uses an approach similar to ours for guiding diffusion models with an unnoised CLIP model. Thank you for bringing this open-source project to our attention. We will cite the suggested work and discuss their work.
>
> ### RE: Experimental Setup - MEIs
> For the GA optimization, we use the established method for generating MEIs that has been tested in vivo [1]. However, we perform a comparison study to show that the parameters chosen are selected to maximize the performance of the GA method. We rerun the MEI optimizations using the AdamW optimizer and find a significant decrease in performance in comparison to the SGD optimizer (r = 0.69). We also run the MEIs for 100 steps instead of 1000 and also find a performance decrease (r = 0.95) (Fig. F).
>
> ### RE: Norm constraint
> Controlling for the norm/contrast of the image is a standard procedure when optimizing MEIs [2] because neurons are strongly driven by contrast. When the contrast is not controlled a trivial solution is to simply increase the contrast of images to values that are not realizable by a monitor. Furthermore, controlling the norm also controls the locality of the synthesized images. When optimizing MEIs, we choose a lower norm because neurons have a localized receptive field, for MENIs we increase the norm to 50, to allow for more full-field images, for the reconstructions, we further increase the norm budget to operate in a similar contrast domain as the ground truth natural images.
>
> ### RE: Evaluation
> The goal of MEI generation is to elicit maximal responses in the brain. Most exciting natural images (selected from the ImageNet dataset) have been established to be less activating than the GA-optimized MEIs [1], which have also been tested in vivo. Therefore, a ground truth image to which MEIs could be compared via standard computer vision metrics does not exist. Hence, for MEIs, we rely on the predicted neural responses as the established metric of MEI performance. Since validating the images in the recorded neurons is not viable for us we introduced the cross-architecture evaluation paradigm as the *in silico* proxy evaluation of the stimuli transferability to the brain. We quantitatively show that the EGG MEIs are more robust to the model idiosyncrasies and we would thus expect them to perform better *in vivo*. Another reason why we use neural activation is that, for the reconstruction of images from neural responses, it has been previously shown that standard computer vision metrics do not necessarily appropriately capture the desired objective i.e. better correlation of the neural responses of the reconstructed image with the responses to the real image [3].
>
> We agree, however, that additional metrics could be helpful for the MENIs and Reconstructions. Please see the general response for the additional evaluation results and discussion.
>
> ### RE Q2: Are EGG MEIs actually better?
> For MEIs the hallmark metric is their performance in terms of maximizing neural responses. Natural images, on the other hand, are not the ground truth for MEIs. As seen in [1, 4], existing MEIs outperform natural images in the brain. Therefore, comparing MEIs to natural images via SSIM/Perceptual Loss would not indicate improvement in maximizing in-vivo neural responses. Please let us know if we somehow misunderstood your point.
>
> ### RE Q3: MENIs distributional similarities
> We performed the requested evaluation of distributional similarities (see Fig. D and general response).
>
> ### RE Q4: Base model clarification
> Firstly, we will improve the consistency of model naming: i.e. *Gaussian* is the ResNet + Gaussian Readout model, *Attention* is the CNN + Attention Readout model.
>
> Table 1: The models in the brackets are the base models, for *within* we use the same architecture, for *cross* we use the other architecture. We will clarify that in the caption of the table.
>
> Figure 3B: The label to the left of the plots is the base model, i.e. blue is the *Gaussian* model (ResNet + Gaussian readout), and pink is the *Attention* model (CNN + Attention readout). We will make that clearer in the caption of the figure.
>
> Figure 5C: We use the *Gaussian* model, we have also included the *Attention* model comparison in the rebuttal. We will make that clearer in the caption.
>
> Figure 6B: We use the *Gaussian* model, we have also included results from the *Attention* model in the rebuttal. We will make this clearer in the caption.
>
> ### RE Q5: Solver
> We chose to use DDPM over the other solvers for its simplicity and closest resemblance to Langevin Dynamics providing the most elegant framework for incorporating energy gradients. It is possible that even better results could be achieved with more complex solvers. However, we considered this out of scope for the current study.
>
> **References**
>
> [1] Walker et al “Inception loops discover what excites neurons most using deep predictive models” (2019)
>
> [2] Willeke et al. “Deep learning-driven characterization of single cell tuning in primate visual area V4 unveils topological organization” (2023)
>
> [3] Cobos et al. “It takes neurons to understand neurons: Digital twins of visual cortex synthesize neural metamers” (2022)
>
> [4] Bashivan et al. "Neural population control via deepimage synthesis" (2019)

---

> > ### Comment · Reviewer_H4Az · 2023-08-11
> >
> > After careful consideration, I have bumped the score from a 3 (reject) to a 4 (borderline reject).
> >
> > I think while the paper proposes an interesting method, there are still a couple of aspects that give me pause. **Considered as a whole, I think there needs to be substantial revisions to the original paper to clarify the scope of the original work.** I understand that revisions cannot be submitted at the current stage, and my comment is more regarding the scope of the needed changes.
> >
> > I thank the authors for their extensive and clear response. I have re-read the main paper, the supplemental materials, the general response, and the subject-wise responses.
> >
> > * It is slightly concerning to me the authors did not initially cite prior work on using the derivative of an energy function to guide diffusion, or were not familiar with widespread use of the predicted x0 trick (both in crowsonkb's work, and the diffusers' library). I understand that revisions cannot be submitted during this period. On re-reading the original paper, **I still get the sense that the authors were overly broad in their claims, even after reading the rebuttal.**
> > * I don't completely buy the claim that a norm constraint is needed, as current energy guided models (via CLIP) do not, and neither do approaches which use diffusion + energy to do image colorization, or inpainting, or object replacement. **This suggests to me that perhaps some aspect of their system is not well tuned.** In theory, the diffusion model should yield a constraint automatically that pulls the generated image towards the distribution of natural images, and the use of a norm constraint suggests that perhaps there needs to be more tuning. I understand the claim by the authors that a norm constraint is standard practice, but the paper you cite also doesn't use a diffusion model.
> >
> > On the other hand, the following responses have answered my questions:
> > * On the prior work side, I agree that GLIDE & DALLE-2 do not use the predicted x_0 trick for gradient guidance. My point was confusingly worded, and I meant to indicate that GLIDE & DALLE-2 use CLIP/energy based gradient conditioning with noisy images.
> > * On additional FID scores, SSIM scores, and VGG scores, I thank the authors for proving those values in the new PDF. I would have been happy with any one of the scores (or CLIP distance as in most diffusion work! But it struck me that my request for CLIP was perhaps not totally valid, as CLIP eval is typically applied to naturalistic RGB images, while the authors focus on greyscale images, so SSIM/FID scores are more appropriate.)
> > * I agree the solver is not in the scope of this study, I meant it more as a suggestion for potential improvements. This work is one if the only papers that I've read on diffusion models in the last two years that uses the original DDPM setup, and not a fancy DDIM/PNDM/DPM-Solver (or the brand new UniPC solver).

---

> > > ### Author Response · Authors · 2023-08-14
> > > **Response to Reviewer H4Az**
> > >
> > > Thank you for your quick response and increasing the score. We would like to respond briefly to your two remaining points.
> > >
> > > ### RE: Scope
> > > We would like to respectfully disagree. We did cite prior work for both in the original manuscript. For the clean sample trick, we wrote: “This is achieved by a simple trick, used in the code of Dhariwal and Nichol [46], of inverting the forward diffusion process” (l.163)
> > >
> > > Regarding the gradient guidance we wrote “Here we extend this [Dhariwal and Nichol et al. 2021] approach to i) use neuronal encoding models, such as the ones described above, to guide the diffusion process and ii) to use a model trained on clean samples only.” (l.153), thus saying that we extend it for neural encoding models, not generally.
> > >
> > > So while we did not cite all references you suggested (thanks for pointing them out), we neither claimed originality on the clean sample trick nor guidance. We understand that it is an important issue and we will make sure to be even more clear about this point in the revised manuscript.
> > >
> > > ### RE: Norm Constraint
> > > We understand now where your concern comes from, but just to re-emphasize: The motivation behind the norm constraint comes from neurophysiological experiments. This constraint aims to prevent confounding results, as actual neurons respond to contrast [see, e.g. Cheng et al. 1994]. Specifically, when comparing two images by how much they activate a neuron in an experiment, both images need to have the same norm/contrast to avoid a trivial result for which one image just has more contrast than the other. When you check experimental papers that use MEIs, you will find that they control for either the contrast or the norm. For instance, [Walker, Sinz et al. 2019 - Nature Neuroscience] rescale the GA MEI after generation to a particular contrast level [Franke, Willeke et al. 2022 - Nature, Bashivan et al. 2019 - Science] directly apply the norm constraint, as we do.
> > >
> > > To show the importance of constraining the norm, we optimized the MEIs without the norm constraint and controlled for contrast post-optimization. We found a decrease to 0.45 of responses to the norm-constrained optimized MEIs in the within architecture validation and 0.81 in the cross validation. Without constraining the norm, the generated MEIs have a mean norm of 1524 and standard deviation of 1511.
> > >
> > > Given the nature of the optimization problem it is expected to observe an escalation of the contrast/norm. For simplicity, imagine a linear neuron model. In this case, the energy gradient leads to a consistent shift towards images with higher norms. This behavior holds true for more complex encoding models as well, since the encoding model's relationship with image contrast is monotonous. Consequently, by introducing the norm constraint, we introduce an additional prior that counteracts the continuous shift.
> > >
> > > This is significantly different from the clip guided methods, while we are looking for a method that provides more meaningful insight into the function of neurons, CLIP guided methods optimize for better looking images. To achieve this they minimize the distance between clip embeddings. The CLIP embeddings cannot be "hacked" by increasing contrast (unlike neurons) and as a result the CLIP-based energy function is robust to contrast and does not require norm constraining, nor is it a requirement posed by the task that these methods are designed.
> > >
> > > **References**
> > >
> > > Cheng et al. “Comparison of neuronal selectivity for stimulus speed, length, and contrast in the prestriate visual cortical areas V4 and MT of the macaque monkey” (1994)
> > >
> > > Walker, Sinz et al. “Inception loops discover what excites neurons most using deep predictive models” (2019)
> > >
> > > Franke, Willeke et al. “State-dependent pupil dilation rapidly shifts visual feature selectivity” (2022)
> > >
> > > Bashivan et al. "Neural population control via deepimage synthesis" (2019)

---

> > > > ### Comment · Reviewer_H4Az · 2023-08-14
> > > >
> > > > I appreciate the response.
> > > >
> > > > I agree that in a unbounded prior-free optimization environment where the neuron activity is monotonic with regard to contrast, optimizing the neuron activity should yield an image with infinite contrast. However, here you are proposing to constrain the synthesis process using a natural image prior, which in the paper you implement using a diffusion process. The diffusion model predicts a (scaled) score function of the image distribution.
> > > >
> > > > However, I do not agree that the same problems in prior-free optimization should necessarily apply to your work (or at least it shouldn't). In my view, by manually applying this scaling, you are changing the synthesized image in a way that explicitly breaks the idea of combining a natural image prior with the neuron prior (as interpreted as a energy guidance anyways).
> > > >
> > > > **I don't consider this theoretically valid. I do not agree that you can dissociate the "texture" of the image (which you keep after rescaling the norm) from the "norm" of the image (which you do not keep), and keep the theoretical interpretation of combining the score with the derivative of an energy prediction.**
> > > >
> > > > Taking into account the your most recent response, I have decided to stand by the current evaluation of this work.

---

> > > > > ### Author Response · Authors · 2023-08-18
> > > > > **Response to Reviewer H4Az**
> > > > >
> > > > > We think we now understand your concern better and believe it might be based on a misunderstanding.
> > > > >
> > > > > ### Unbounded guidance of diffusion models
> > > > > To illustrate that this is not purely the case for prior-free optimization consider a simplified case with a Gaussian prior (representing the natural image prior) and a log-linear likelihood (representing maximizing the neural response by increasing contrast). Individually the scores of the components are $\nabla_x \log p(x) \propto -x$ and $\nabla_x \log p(c | x) = c$, where $c$ is some constant. Thus, if an image is generated from just the prior, then the contrast would be within the typical natural image contrasts. If the image is generated from only the likelihood, the contrast is unbound and will be pushed to $\infty$. Combining the score functions provides the score of the posterior distribution expressed as $s(x) = -x + \lambda c$, where $\lambda$ is the energy scale. It can be seen from this that the mean of the score function is defined by $x = \lambda c$. So for low energy scales, the contrast will remain close to the natural image prior, however, for high values of $\lambda$ (like in the MEIs case) the likelihood will push the contrast to higher values. We observe exactly this effect, when generating MEIs with $\lambda = 1$ the norms are closer to the natural image norms (Testset ImageNet norms: mean 90, std 23; MEI norms: mean 79, std 18); however, for $\lambda = 10$ the norms are much larger (mean 423 and std 501, for the 100x100 resolution).
> > > > >
> > > > > ### Norm constraint is applied in the energy function only
> > > > > We realized that our statement “we set the norm of the 100 x 100 image to a fixed value of 25.” (l.230) might have caused a bit of confusion. The norm constraint is applied **only** before showing the image to the encoding model as a preprocessing step and it is not kept between diffusion steps. Therefore the constraint only affects the generation process through the gradient of the energy function **not** by changing the image directly.
> > > > >
> > > > > We hope this helped to clarify this issue. We will make sure to be more clear about the above points in the text.

---

> > > > > > ### Comment · Reviewer_H4Az · 2023-08-18
> > > > > >
> > > > > > To clarify, in this particular example do you mean $\nabla_{x}\log p(x|c) = a$, where $c$ is the condition where you maximize the response of a neuron, $x$ is an image, and $a$ is a constant?

---

> > > > > > > ### Comment · Reviewer_H4Az · 2023-08-18
> > > > > > >
> > > > > > > Ah never mind, I see now, $c$ is the maximizing condition, $x$ is the image, and the score is proportional to some constant.
> > > > > > >
> > > > > > > In this 1D case, I don't agree there exists a well posed joint distribution at all. Under the second condition, where the score is basically constant w.r.t. the input (image), then this distribution must have infinite mean, and infinite variance (you can't have a distribution with infinite mean and finite variance if we define it using a typical definition, ). What does it actually mean to have a joint distribution in this case? Assuming the prior for natural images has finite mean and variance, then the joint distribution must also have infinite mean and variance.
> > > > > > >
> > > > > > > This in my opinion would not be a well posed joint distribution, and it is not something you can sample from using your typical composition of scores (or anything proportional to the score if we use a relaxed understanding of "joint distribution"). And it is not something that you can sample from using MCMC/langevian dynamics.
> > > > > > >
> > > > > > > I understand you are not modifying the diffusion output at each step, and my objection comes from having to apply any constraints that you define by hand.
> > > > > > >
> > > > > > > In my view, by normalizing the image you are no longer sampling from the joint distribution, but some transform of the sample. This transform is something you define, and it weakens the validity of this method in my opinion.

---

> > > > > > > > ### Author Response · Authors · 2023-08-19
> > > > > > > > **Response to Reviewer H4Az**
> > > > > > > >
> > > > > > > > To be honest, we are not quite sure what it exactly is that you disagree with. Could you please tell us what claims/statements we make that you think are incorrect? If we make an incorrect or imprecise statement, we are happy to correct it or clarify it.
> > > > > > > >
> > > > > > > > ### Re: Clarification
> > > > > > > > In our previous example $E(x) = \lambda c x$, $c$ was indeed the “maximizing condition”, i.e. a linear function that describes the neuronal response to an image $x$. The notation $p(c|x)$ was possibly a bit misleading. We apologize for the confusion this might have caused.
> > > > > > > >
> > > > > > > > ### Re: The “joint” distribution exists and is a proper distribution
> > > > > > > > Your statement “this distribution must have infinite mean, and infinite variance [...] Assuming the prior for natural images has finite mean and variance, then the joint distribution must also have infinite mean and variance.” is not correct. While the linear energy $c \cdot x$ does not determine a proper distribution, the product of the prior and energy **does** yield a normalizable distribution with moments:
> > > > > > > >
> > > > > > > > $p(x| \text{maximize }cx) = p(x)\exp(c x)$
> > > > > > > >
> > > > > > > > $ \propto \exp(-x^2)\exp(cx)$
> > > > > > > >
> > > > > > > > $= \exp(-x^2  + cx)$
> > > > > > > >
> > > > > > > > By multiplying by the constant $\exp\left(-\frac{c^2}{4}\right)$ we can complete the square
> > > > > > > >
> > > > > > > > $\exp(-x^2  + cx - \frac{c^2}{4}\)$
> > > > > > > >
> > > > > > > > where
> > > > > > > >
> > > > > > > > $-x^2  + cx - \frac{c^2}{4} = -(x - ½ c)^2$
> > > > > > > >
> > > > > > > > yielding  a Gaussian shifted by $½c$.
> > > > > > > >
> > > > > > > > $p(x|\text{maximize } cx) \propto \exp(-(x - ½c)^2)$.
> > > > > > > >
> > > > > > > > So while the energy is not normalizable the product with the prior is. This also extends to 2D.
> > > > > > > >
> > > > > > > > $p(x |\text{maximize } c^T x)  \propto \exp(-x^T x)\exp(x^T c)$
> > > > > > > >
> > > > > > > > $= \exp(-x^T x  + x^T c)$
> > > > > > > >
> > > > > > > > $= \exp(-x^T x  + x^T c - ¼ c^Tc)$
> > > > > > > >
> > > > > > > > $= \exp(-(x - ½c)^T(x - ½c))$
> > > > > > > >
> > > > > > > > Also note, that it might be misleading to think of it as a joint distribution because $\exp(c^T x)$ is not a likelihood (at least not for the neuronal response).
> > > > > > > >
> > > > > > > > ### Re: the summation of the scores
> > > > > > > > However, not every product of prior and energy function will be normalizable. In that case, it loses the interpretation of a proper posterior. If this is what you take issue with, we are happy to carefully point this out in the manuscript and mention it in the limitations. However, even in that case, we can still use Langevin dynamics to generate an MEI regularized by the prior. Sampling from unnormalized/improper distributions is the point behind Langevin dynamics, which was originally defined for particles moving in an energy field [1]. Described by the Langevin equation the velocity of a particle is $\lambda \dot{x} = -\nabla_x V(x) + \eta$, where $x$ is the position of the particle, $V(x)$ is the energy at position $x$ and $\eta$ is Gaussian noise. In the standard definition $V(x)$ is not necessarily a distribution, but any differentiable function. What can be seen by solving the Fokker-Planck equation for the Langevin equation is that for $V(x) = -\log p(x)$ the steady-state of the stochastic differential equation results in sampling from the distribution $p(x)$.
> > > > > > > >
> > > > > > > > ### Re: Norm Constraint is a neuroscience specific constraint.
> > > > > > > > Again we respectfully disagree with your statement “by normalizing the image you are no longer sampling from the joint distribution, but some transform of the sample.” We must emphasize that the constraint is part of the **energy** function $E(x) = f\left(\frac{x}{||x||_2} n\right)$, where $f$ is the encoding model, $x$ is the image and $n$ is the target norm. The constraint is a component of the encoding model, **not** the generation process. Since we are free to pick whatever energy function is useful, we define the guiding energy to be meaningful to the task we are solving. This choice is dictated by the nature of the neuroscientific problem we are solving (motivated by multiple neuroscience publications mentioned above), and not necessary for the successful functioning of our generation method. In the previous response we showed that our method can generate MEIs without the norm constraint (so the sampling works), but – when normalized to the neuroscientifically motivated norm – does not drive a neuron as strongly as an image generated with an energy that includes this target norm. When we use the energy that includes the normalization, we sample a non-normalized image from the prior times energy that – when normalized to a neuroscientifically motivated norm – drives a selected neuron strongly.
> > > > > > > >
> > > > > > > > **References**
> > > > > > > >
> > > > > > > > [1] P. Langevin “‘Sur la théorie du mouvement brownien” (1908)

---

> > > > > > > > > ### Comment · Reviewer_YaDz · 2023-08-19
> > > > > > > > > **Question from another reviewer about this thread**
> > > > > > > > >
> > > > > > > > > >In the previous response we showed that our method can generate MEIs without the norm constraint (so the sampling works), but – when normalized to the neuroscientifically motivated norm – does not drive a neuron as strongly as an image generated with an energy that includes this target norm. When we use the energy that includes the normalization, we sample a non-normalized image from the prior times energy that – when normalized to a neuroscientifically motivated norm – drives a selected neuron strongly.
> > > > > > > > >
> > > > > > > > > I just want to understand something correctly what you mean by "does not drive a neuron as strongly". Did you actually confirm (via recording from neurons) that this is the case? Or is this based on the model responses, which could have specific biases?
> > > > > > > > >
> > > > > > > > > Just as a note of agreement with reviewer H4Az, as the authors mentioned, by adding in this norm-constraint it is imposing a separate prior *in addition* to that imposed by the diffusion model. I have generally found the addition of these "ad-hoc" priors like norm-constraints or TV regularization unsatisfying in previous work (which was why I was excited about this because it seems more principled and based on the statistics of natural images!) But unfortunately it seems like you still need some of these additional constraints in addition to the prior from the diffusion model, which makes things a bit less satisfying, and so I tend to agree that this weakens the validity of the method. It seems like there should be a way that you can incorporate this into what the diffusion model learns from the start.

---

> > > > > > > > > > ### Author Response · Authors · 2023-08-20
> > > > > > > > > > **Response to Reviewer YaDz**
> > > > > > > > > >
> > > > > > > > > > Thank you for your response. Please find our response to your points below.
> > > > > > > > > >
> > > > > > > > > > ### Re: Does not drive a neuron as strongly
> > > > > > > > > > The paper does not include neurophysiological verification experiments. So this is based on the  model response which could have biases. Sorry, if that was unclear.
> > > > > > > > > >
> > > > > > > > > > Additionally, it is important to note that the unconstrained MEI does not drive the model neuron as strongly if the images are compared at the same norm. However, if it is not constrained to the same norm it does drive the model neurons stronger (1.81 times; please also see our explanation below).
> > > > > > > > > >
> > > > > > > > > > ### Re: Prior by the diffusion model
> > > > > > > > > > We try to distinguish a few different components which hopefully makes things clearer:
> > > > > > > > > > - Biological neurons are very strongly driven by contrast. The encoding model learns this very easily. So when optimizing MEIs without norm constraint the images usually strongly grow in contrast, possibly even outside the range of what can be shown in an experiment with a monitor. This is **independent** of the optimization method used, i.e. it’s also true for straightforward gradient ascent (GA). Of course, real neurons will saturate at some point, but this saturation point might be outside the responses that the encoder gets to see from the recorded images. Thus, this saturation point could be out-of-domain for the encoding model.
> > > > > > > > > > - Since the contrast direction is strong and “neuroscientifically trivial”, images that are presented in biological experiments are contrast/norm normalized to make the responses between different presented images comparable (see e.g. Walker et al. 2019 or Willeke et al. 2022). A particular target norm is usually achieved by simply rescaling the image to that norm.
> > > > > > > > > > - As reviewer H4Az noted, the norm and content of an image might not be independent for a neuron. Thus we want to optimize an image that, **when rescaled to the target norm chosen by the experimenter**, yields a strong response. This is the reason for the norm constraint. Optimizing an image without the norm constraint and then normalizing it before showing it to a neuron (here model-neuron, but could be biological neuron) might yield a suboptimal response (as we showed for a model-neuron before), because that actually assumes independence of norm and content i.e. that changing the contrast with the same texture would have no influence on performance. So the norm constraint in the energy function just encodes at which norm we ask the encoder whether it likes the MEI, because this is what we want to end up with.
> > > > > > > > > > - Most importantly, however, it is not needed to generate images, i.e. it’s not a necessary regularization hack. The images that are generated without norm constraint do drive the neurons very strongly, **also stronger than the images with norm constraint (1.81 times stronger)**. This is not surprising because the MEI method exploits the neuron’s preference for contrast and gives it a high norm – usually higher than the norm of the norm constraint. It only doesn’t drive the (model)-neuron as well as the norm constrained MEI when **rescaled back to the same norm**. To put it succinctly (let uMEI be the MEI found without norm constraint and nMEI with norm constraint)
> > > > > > > > > >   * encoder(uMEI) > encoder(normalize(nMEI))
> > > > > > > > > >   * encoder(normalize(nMEI)) > encoder(normalize(uMEI))
> > > > > > > > > > - Finally, it is worth mentioning that  even OpenAI in their Guided Diffusion code from "Diffusion Models beat GANs" clip the images to the correct image range (scripts/image_sample.py l.60, https://github.com/openai/guided-diffusion/blob/22e0df8183507e13a7813f8d38d51b072ca1e67c/scripts/image_sample.py#L60). The same is true for Stable Diffusion (https://github.com/huggingface/diffusers/blob/74d902eb59f873b6156621220937f8e2521dfdc0/src/diffusers/image_processor.py#L117C17-L117C17). So none of the diffusion models actually guarantee that the image is within the correct value range. Since the encoder cannot have seen images beyond the normal 8-bit range, because they cannot have been presented in an experiment, images outside that range will also likely be out-of-domain for this model. This is another good reason to constrain the values of the generated images, independent of any neuroscientific or potential regularization criteria.
> > > > > > > > > >
> > > > > > > > > > To summarize, we can generate images without norm constraint, and they do drive neurons 1.81 stronger than the images generated with norm constraint, but they have a very strong “contrast component” introduced by the encoding model’s learned preference of real neurons for contrast. However, this component is neuroscientifically less interesting so it’s removed.
> > > > > > > > > >
> > > > > > > > > > We hope this clears things up. If not, please let us know.

---

> > > > > > > > > > > ### Comment · Reviewer_H4Az · 2023-08-21
> > > > > > > > > > >
> > > > > > > > > > > Just a quick note.
> > > > > > > > > > >
> > > > > > > > > > > - As a point of practicality, I think a more sound proof can be achieved by upper bounding the response of an encoder. For example: $\text{pred} = \min(\text{encoder}(image), 2\times\text{neuron}_\text{max})$, you could do this for both the minimum and maximum of a neuron using the actual observed min and max of a neuron to natural images, and here I choose 2 arbitrarily. In this case, the probability density function would be guaranteed to be bounded to a finite interval, and in my opinion would solve the problem of the energy function having an unknown (potentially infinite) normalizing constant.
> > > > > > > > > > > - For norm and contrast, my point is that **for natural images** I do not think you can separate the norm and the content. By rescaling the input at the end, I don't think you are sampling from the joint distribution if there was one. **I still dislike your rescaling approach**, and believe your approach has some aspect that is not well tuned (probably overfit encoder). Ideally you could just tune a single step size hyper-parameter (potentially even for each neuron), and this approach would just work.
> > > > > > > > > > > - **I don't believe your norm rescaling is comparable to Guided Diffusion/Diffusers.** With exception of the clamping operation, their rescaling is the same affine operation for every image and even arbitrary sets of images. Your norm rescaling is not the same affine operation for every image, and would differ for images with different norm. **In my opinion, your approach is weakened by introducing this norm rescaling operation, as it is a somewhat hand derived "prior".**
> > > > > > > > > > >
> > > > > > > > > > > That said, I do hope the authors at least acknowledge this as a potential weakness/flaw in their paper.

---

> > > > > > > > > > > > ### Author Response · Authors · 2023-08-21
> > > > > > > > > > > > **Response to Reviewer H4Az**
> > > > > > > > > > > >
> > > > > > > > > > > > Thank you for increasing your score. We are happy to include the discussion on the norm constraint in the text and acknowledge it as a potential limitation.
> > > > > > > > > > > >
> > > > > > > > > > > > We appreciate your engagement and were happy to discuss.

---

> > > > > > > > > > > > > ### Comment · Reviewer_H4Az · 2023-08-21
> > > > > > > > > > > > >
> > > > > > > > > > > > > Yes. Please include at least a brief discussion on the use of hand crafted normalization (probably in the limitations section and future work section) in an upcoming revision.
> > > > > > > > > > > > >
> > > > > > > > > > > > > I do think this is an important flaw to acknowledge, and it weakens the current proposal somewhat.

---

> > > > > > > > > ### Comment · Reviewer_H4Az · 2023-08-19
> > > > > > > > >
> > > > > > > > > After reading your comment, I have raised the score to a 5 from a 4.
> > > > > > > > >
> > > > > > > > > My previous comment wasn't well worded, and I apologize.
> > > > > > > > >
> > > > > > > > > My concern was mainly regarding two aspects:
> > > > > > > > > * Is the joint distribution well posed? In that I mean does it have finite mean and finite variance (as opposed to infinite mean and infinite variance)?
> > > > > > > > > * By applying a manual norm constraint (a "prior" in a sense), you are incorporating knowledge that is outside of the method itself, and if the distribution was indeed well posed, then in this case are you NOT sampling from the distribution? But rather some transform of a sample?
> > > > > > > > >
> > > > > > > > > I agree you are correct regarding the joint distribution. I ignored the possibility of the potential growth/shrinkage of the different distributions as $\lim x \rightarrow \infty$. And that under a normal prior, where the other distribution has PDF proportional to $\exp (c \cdot x)$ (assuming for a second that this was a distribution), then the joint distribution is well posed (finite mean and variance). I was thinking about this just now, and I thought the trivial case would be a distribution where one is non-zero on a finite interval, so your example using the normal distribution with infinite support is good.
> > > > > > > > >
> > > > > > > > > This does not solve the problem if the energy function has a finite normalizing constant, but I guess if you are only sampling from the joint distribution it is fine, as the prior should be sufficiently strong at some point.
> > > > > > > > >
> > > > > > > > > I am still bothered by the use of a manual norm constraint, and I don't quite agree it doesn't matter just because you are applying it to the output. Ideally your method would not need this, as it is something you hand define. It weakens the validity of the approach a bit, and **I encourage the authors to note this explicitly in text as a potential limitation**.
> > > > > > > > >
> > > > > > > > > I appreciate the authors for the engagement and clarifications.

---

### Official Review · Reviewer_GN1s · 2023-06-30

**Soundness:** 3 good
**Presentation:** 3 good
**Contribution:** 3 good
**Rating:** 7
**Confidence:** 3

**Summary:**

Further characterizing the complex coding properties of V4 neurons might require (1) better encoding models of neuronal activity as well as (2) better methods to generate informative most exciting inputs (MEI). The paper tackles (1) by proposing a new readout mechanism for a convolutional data-driven core based on attention that outperforms the SOTA at predicting neural responses of neurons in macaque V4. Then, it tackles (2) through a new simple method called EEG that generates MEI using a pre-trained diffusion model and the guidance signal of the encoding model. EEG can also be used for image reconstruction from neuronal activity, in both applications it generalizes better than the more traditional gradient ascent (GA) across architectures. Finally, their method can produce most exciting natural inputs (MENI) on par with highly activating natural images.

**Strengths:**

- The problems tackled are well motivated
- The paper is clear
- The proposed readout mechanism is: novel to the best of my knowledge, biologically motivated, sound, and backed up by positive results (outperform current SOTA in predicting neuronal activity)
- The method to generate MEI is: ingenious, simple but sound, and backed up by positive results (better generalization than GA and faster)

**Weaknesses:**

- (1) compares two models with: a different core, a different readout, and a different training strategy, hence it is hard to isolate the benefit from the new readout mechanism only. This part would be strengthened with additional baselines like the core model with Gaussian readout and the pre-trained resnet50 with attention readout
- While the new model seems to be a better encoding model of neuronal activity --according to the results of (1)--, MEI generated with it seem overall worse at driving neurons activation (fig 5.b) compared to the baseline ResNet50 (both EGG and GA). In fact, the results on MENI reported in Fig5.c are with the ResNet50 model, and the examples of image reconstruction put forward in the paper all come from the ResNet50 model.

**Questions:**

- Did the author consider performing a CKA between models in the within condition and between models in the cross condition, to quantify the generalization/transferability of the MEIs?
- Why is the ResNet50 and not the ACNN considered for comparing MENI to top1 most activating ImageNet images? At least this is what I am inferring from the color used in Fig 5.c, as it is not written in the text which model is finally used.
- It would be interesting to add GA as a baseline for fig 5.b. I assume that the intuition behind the plateau/decrease is that if we increase the energy scale "too much", we fall back to "overfitting to the idiosyncrasies" of the encoding model, this baseline would give a bit of context w.r.t to this hypothesis.
- How are the images shown in the paper selected?
- My understanding for the motivation behind the MENI proposed is speed. Getting more natural-looking stimuli (e.g., for control stimuli) without having to go through millions of images. While I agree with the motivation, given the random nature of the generative process, I was wondering how much sampling was necessary before falling on satisfactory MENIs? Will I trust that EEG will still be faster, I worry about the meaningfulness of some of the MENIs generated (In Dhariwal et al. 2021, samples generated with a lower guidance scale are of significantly lower quality, e.g., $\lambda=1$ leads to FID=33)
- For image reconstruction, it would be interesting to see examples generated with the ACNN model, only results from ResNet50 are shown or discussed in the paper.



Clarification comments:
- Slight lack of consistency w.r.t to naming: the encoding model is called through the paper: data-driven with attention readout, Attention CNN, ACNN, and Attention.
- Within: l.201: "task-driven ResNet with Gaussian readout or data-driven with attention readout" & Cross: l.203: "ResNet and data-driven with attention readout". I assume in both case the ResNet uses a gaussian readout

**Limitations:**

- It is important to note that, while the trick used to make the classifier guidance works on models not trained on noise does allow for more flexibility, the encoding model still has to be trained on images from the same dataset as the diffusion model, as the approximate clean sample $x_0$ falls within the data distribution of the diffusion model and not necessarily of the encoding model. If not, the $x_0$ will be o.o.d to the encoding model and its gradient will be significantly less meaningful.

---

> ### Author Rebuttal · Authors · 2023-08-08
>
> We would like to thank you for your in-depth review.
>
> Please find our responses to your questions below.
>
> ### RE Ablation study
> See general response.
>
> ### RE: Apparent decrease in driving neural response
> We apologize for the confusion in Fig. 5b. The values there are shown as normalized to the max responses of each neuron across lambdas.  Thus, when normalizing, the values become smaller than 1. The result indicates that there is less consistency in the *Attention* model as to which value of lambda is most activating. We replotted the figure normalizing the neural activation of the GA-optimized MEI (Fig. G) and we will replace Fig. 5b with it.
>
> ### RE: using the Attention Model for reconstructions
> We attach *Attention* model reconstructions (Fig. H), showing that they perform similarly to the *Gaussian* model (Fig. J). We will add these figures to the appendix. We chose to reconstruct using the *Gaussian* model, such that the cross-architecture evaluation is performed on the *Attention* model. This is because the *Attention* model is better correlated with the brain responses and thus provides a better proxy for the brain. Thus, better mimic the setup of testing reconstructions in the brain.
>
> ### RE: Using CKA
> Centered Kernel Alignment compares representational similarity, which would be applicable to the core representations. However, investigating the core representations was not the focus of our paper since we focused on finding the readout that would best predict the neural responses. As we show in the ablation experiment (see our general response), the attention readout shows performance superior to the Gaussian readout even when applied on top of exactly the same core. However, we might have misunderstood your suggestion and are happy to comment if you clarify it.
>
> ### RE: Using *Attention* model for top-1 ImagNet comparison
> We will include the *Attention* model comparison (Fig. I) to the top-1 ImageNet images in Fig. 5. The *Attention* MEIs at $\lambda = 1$ slightly outperform the top-1 ImageNet images.
>
> ### RE: add GA as a baseline for Fig 5.b
> Please see **RE: Apparent decrease in driving neural response**
>
> ### RE: How are the images shown in the paper selected?
> The neurons used for the study are randomly chosen from the neurons with test correlation > 0.5. The images shown in the main text are selected to show the performance of various neuron properties (eyes, fur, edges, curves). The complete set of images is shown in the appendix.
>
> ### RE: motivation behind the MENI and sampling required
> For the MENIs presented we generate them from 3 seeds and choose the highest activating of the 3 seeds. Our main objective was to show that controlling $\lambda$ allows us to move closer to the natural images manifold. We do not consider it a replacement tool for searching natural images, but rather an additional tool for better interpretability of MEIs as at times (e.g. top row ResNet fig.5A) it is difficult to interpret the MEI ($\lambda=10$) and traversing $\lambda$ can help to interpret the function. For FID we include the comparison of FIDs across $\lambda$.
>
> ### RE: Naming consistency
> Thank you for pointing out the inconsistency. We will unify the naming to generally use *Attention model* for the data-driven with attention readout model. For the task-driven ResNet model + Gaussian readout we will refer to it generally as *Gaussian model*. We will also add a statement that defines these two terms to avoid confusion.
>
> ### RE: l.201 and l.203
> As in the previous section, we will unify the naming to remove confusion. Yes, the *task-driven* (*Gaussian*) model always uses the Gaussian readout (except for the new ablation study).
>
> ### RE: Encoding model needs to be trained on the same distribution
> You raise an interesting point. However, we do not entirely agree. Mainly, we consider the problem the other way around: The goal for encoding models is to faithfully represent the visual system as well as possible, no matter what kind of image is shown. The encoding model is mainly determined by the choice of images shown by the experimenters. The diffusion model, on the other hand, determines the space (prior) of images within which the most exciting images are generated. For example, it would be possible to use a diffusion model trained on sketches to generate the most exciting images in the space of sketches. Since the visual system has evolved to encode natural images, we focus on diffusion models for natural images here.
> We will include this discussion in the Discussion section.

---

> > ### Comment · Reviewer_GN1s · 2023-08-14
> >
> > The authors have successfully addressed my 2 main concerns, hence I upgrade my update my rating from 6 to 7.
> > I will also reply to the few minor concerns left below:
> >
> >
> > ### RE: Apparent decrease in driving neural response
> >
> > Thank you for clarifying figure 5b. I agree that figure G better communicates that MEIs generated from the Attention model are better drivers of neural response than the one from the ResNet model. That being said, figure 5b highlights the fact that MEI generated with EEG for different neurons can be pretty sensitive to the energy scale hyperparameter, which is the case with the Attention model but not with the ResNet model. Hence, I believe that this figure still has its place in the paper to discuss such limitations of the method, albeit in the appendix.
> > Do the authors have a hypothesis as to why their model is more sensitive to the energy scale?
> >
> >
> > ### RE: Using CKA
> >
> > I will clarify my question. The paper cares to evaluate the generalizability of the MEI generated from one architecture to another as a proxy for its application on the brain. To quantify how much downstream generalizability we can expect, it seems reasonable and informative to perform a CKA between the representations (which are driving the generation of MEIs through the gradient) of the 2 architectures tested.
> >
> > ### RE: Encoding model needs to be trained on the same distribution
> >
> > While I understand the logic in theory, in practice, if the images used to train the diffusion model and the ones used to train the encoded are too dissimilar, I am not too clear how meaningful the gradient from the encoder will be to drive the generation of the MEIs.

---

> > > ### Author Response · Authors · 2023-08-18
> > > **Response to Reviewer GN1s**
> > >
> > > Thank you for increasing your score. We are happy to see that we have addressed your main concerns. We would like to briefly address your remaining concerns.
> > >
> > > ### RE: Attention model is more sensitive to the energy scale
> > > The Attention data-driven model is not pretrained on natural images. We thus hypothesize that it has less of the natural image bias than the task-driven Gaussian model. This can be seen when comparing the different MEIs:
> > >
> > > - **Attention Model GA vs Gaussian Model GA**: GA MEIs for the Attention model retain less of the naturalistic features (e.g. eye neuron column 5 Fig. 3a).
> > > - **Attention Model EGG vs Gaussian Model MEIs**: The EGG generated MEIs show more naturalistic features similar to the MEIs obtained for the Gaussian Model.
> > > - **Attention Model EGG, different lambdas (5 & 10)**: By increasing the lambda from 5 to 10 (less naturalness) we observe MEIs that are more similar to the Attention Model GA MEIs. We will include examples of Attention model MEIs with lambda 10 in the appendix in the revised manuscript.
> > >
> > > We hypothesize that, since the data-driven model is trained on neuronal images directly, it has a smaller bias towards natural images which allows the model to better predict neural responses in the Attention model. However, this could make it more difficult to generate generalizable MEIs, because the optimization might overfit to the idiosyncrasies of the model. Therefore, by using EGG we can control the amount of naturalness bias that the MEIs have.
> > >
> > > ### RE: CKA
> > > We now better understand your point with CKA. We computed CKA of the neural encodings *across* architectures between the Attention model and Gaussian model and *within* architecture between different seeds (e.g. Attention 1 and Attention 2 are models with the same architecture, but trained with different seeds). The CKA is computed between the predicted neuronal responses.
> > >
> > > | Model       | Attention 1 | Attention 2 | Gaussian 1 | Gaussian 2 |
> > > | ----------- | ----------- | ----------- | ---------- | ---------- |
> > > | Attention 1 | 1           | 0.9949      | 0.9133     | 0.9116     |
> > > | Attention 2 | 0.9949      | 1           | 0.9145     | 0.9129     |
> > > | Gaussian 1  | 0.9133      | 0.9145      | 1          | 0.9994     |
> > > | Gaussian 2  | 0.9116      | 0.9129      | 0.9994     | 1          |
> > >
> > > We observe that the within architecture similarity is very high (> 0.99) for both architectures and the cross architecture similarity is slightly lower, but also high (> 0.9). We expect such an outcome, since both architectures were trained to model the same neural representation.
> > >
> > > ### RE: Same training dataset between encoding model and diffusion model.
> > >
> > > We agree that in practice the encoding model needs to generalize to the manifold of the diffusion model. Prior work has shown that in mice encoding models do generalize to some extent outside their training manifold [1]. However, how far they generalize is an empirical question. Nevertheless, we will add the discussion point, that “the encoding model needs to generalize to the manifold of the diffusion model” to the limitations.
> > >
> > > **References**
> > >
> > > [1] Wang et al. “Towards a Foundation Model of the Mouse Visual Cortex” (2023) bioRxiv

---

### Official Review · Reviewer_YaDz · 2023-07-06

**Soundness:** 3 good
**Presentation:** 3 good
**Contribution:** 3 good
**Rating:** 6
**Confidence:** 4

**Summary:**

This paper proposes using the prior implicit in a diffusion model as a regularization term when generating images that maximally excite neurons. The authors refer to this as “Energy Guided Diffusion” and compare this to a standard gradient assent procedure with a smoothness prior.  The authors also introduce a new model consisting of a CNN with an attentional readout that is trained to predict the responses of biological neurons, which (unlike the comparison model) allows for spatial components to be weighted differently for each input stimulus. The authors conduct experiments on all combinations of these two model changes to see how the generated stimuli change, and analyze how the stimuli transfer across models.

**Strengths:**

Overall, I enjoyed this paper and found it accessible, interesting and a good combination of many ideas that are currently relevant for the NeurIPS community.
* The authors incorporation of an attention module into the final layer of a model for neural predictivity is novel as a way to capture stimulus-dependent changes
* Showing the MEIs replicate for the chosen neurons and have very similar properties even with different models (ie  Figure 3) is a nice validation of the MEI technique as way of interpreting neural tuning
* I was excited to see the prior in a diffusion as a way to improve some of the neural synthesis techniques, as this is perhaps a bit more principled than some previous methods (i.e. smoothness priors, or GANs).


**Weaknesses:**

Overall, I think that the contribution of the paper is novel, however there are some weaknesses in the experimental design and the interpretation/presentation of the results.

A) It is not possible to determine which change in the proposed model with an attention readout leads to better predictivity in Figure 2. This makes the rest of the results comparing the models difficult to interpret. Specifically, compared to the “Task-Driven” ResNet, the authors have changed (1) the core model architecture from a ResNet50 to a 4 layer CNN (2) the readout from the “Gaussian” readout to an attention readout and (3) the training task/dataset (the “task-driven” core of the ResNet50 is pre-trained on image recognition, while the entirety of the “data-driven” attention CNN is trained explicitly to predict the responses of V4 neurons). Which of these changes are critical for the performance increase? Are the changes due to the utilization of the attention readout, the fact that more parameters can be learned in the attention CNN, or some other change?

B) Along the lines of the above, the authors talk about the attention change as being novel and providing a way to incorporate shifts in location of receptive fields, but there is no experiment showing that the learned attention readout *actually* learns to do this (it could be using the same location for every image).

C) Perhaps my biggest concern – the authors discuss generating “most naturally exciting images” however the generated images are not natural. Calling these images “natural” is incredibly misleading, as there is an underlying diffusion model that has learned a specific distribution of image statistics. The generated images may be high-probability images given the prior learned by that diffusion model, but unless the diffusion model is a perfect model of the world these are not fully “natural” (and indeed, from the presented images they do not look natural). It is interesting to present images with different regularizations applied, however, in my opinion, these images should not be presented as “natural” and fully removing the paragraph on lines 262-268 (and also any associated references to “MENIs”) would only strengthen the paper.

D) In the discussion the authors note that the MEIs generated for the attention-readout model are more “concentrated” however, as far as I can tell, this seems to be just an observation and not quantified in any way.


**Questions:**

Some of my major questions were listed in the weaknesses section, but additional things would improve the paper.

1)	In Figure 5b, I’m a little confused about how the increase in the energy scale (so a higher importance on the “maximization” component of the synthesis) starts to decrease for large values of lambda. Naively, it seems like this component of the loss should only get better when it is weighted more strongly. Is there a problem with the optimization (i.e. too large of steps from the energy gradients?) causing this decrease?

2)	The authors should potentially discuss other ways that people have “regularized” generated images and the limitations. For instance, in Bashivan, Kar, DiCarlo (2019) a TV loss was incorporated into the gradient assent procedure to encourage smoothness (similar to the gaussian blur on the gradients used here). Some of these other methods have directly trained models that incorporate some statistics of the “natural” world, such as GANs (Ponce et al. 2019 and follow-up work). Limitations of including priors into synthesis techniques have been discussed in Engstrom et al. 2019 and Feather et al. 2022 (specifically, the inclusion of a prior for generation can hide some of the model biases).

3)	With regard to the limitations of priors mentioned above, the authors mention that the MEIs “look more complex and natural” and that EGG improves the “quality” of the generated images. Can we gain understanding from having things that “look” better? Or are these potentially just misleading us to finding things that *seem* interpretable? (This is maybe a bit more of a philosophical question beyond the scope of the work, but it is something I am sometimes puzzled by with regard to these neural generation procedures, especially when neural data on the generated stimuli is not present).

4)	What sort of biases do we expect from the fact that these models are trained to only handle clean images? The authors state as a fact that neural encoding models are trained on responses to “clean” images, but this seems like it would bias the generation in specific ways.

5)	How much spread is there in the generation across multiple seeds? Specifically, how does the spread in Figure 3 compare to what is observed from multiple initializations within the same model?

Minor clarifications:

a)	What are the training details for the pre-trained robust ResNet50 that is used (ie $l_p$ norm, $\epsilon$ step etc)? And are multiple pre-trained versions used in the ensemble or is it just a different readout?

b)	How is the feature space for this model 1024 dimensional? Are the activations spatially averaged?

c)	The paper “Solving linear inverse problems using the prior explicit in a denoiser” by Zadkhodaie and Simoncelli came to mind as very relevant previous work, but I didn’t see it cited.

d)	It might be helpful to have an additional line/equation around line 166 explicitly stating the generation steps in terms of $\bar{x}_0$.

e)	In Figure 6 it states that the distance is compared in “unit activations space”. Is this the same as the predicted neural response, or is it measured at an internal model stage?

Typos/confusions:

i)	Line 160 says that a constant value of lambda is used for the study, however this was searched over (I think) based on Figure 5?

ii)	Should equation (5) and associated text be $\epsilon_\theta$?


**Limitations:**

The authors have a section on limitations, however there are a few things along these lines that might be worth further addressing.
* The authors do not validate the generated stimuli with neural recordings and the improvement in “quality” is only judged qualitatively or by looking at the predicted responses from other models. It is possible the generated stimuli with the diffusion generation or the attention model do not perform as well at controlling the neural data due to underlying biases in the sampling etc.  I understand that this would require additional neural experiments, however it is worth mentioning this as a limitation and I didn’t see it discussed.
* Are there possible negative societal impacts of the work?

---

> ### Author Rebuttal · Authors · 2023-08-08
>
> We would like to thank you for your in-depth review and we are glad you enjoyed the paper.
>
> Please find the responses to your questions below:
>
> ### RE A: Identifying what contributes to the improved performance
> We performed an ablation study showing that the *Attention readout* is critical for improving performance. For more details please see the general response.
> ### RE B: Experiment to show that the attention readout exhibits stimuli-driven receptive field shifts
> We analyzed the attention maps (i.e. receptive fields) of the attention model. We show that across the test stimuli for the majority of units the receptive field shifts to different locations (Fig. E). See the general response for details.
> ### RE C: Generated "Natural" Images
> We realize that there might be different definitions of what constitutes a natural image. To avoid any confusion we will refrain from calling the images “natural”. We will move the top-1 Imagenet comparison to the appendix, and we will show that decreasing $\lambda$ results in lower FID with the top-5 ImageNet dataset (Fig. D), indicating that the images become more similar to natural images and thus are close to the "naturalness" manifold.
> ### RE D: Concentrated Attention MEIs
> To show that the MEIs from the *Attention* model are more concentrated we compute an isotropic Gaussian envelope for the MEIs. We find that the *Attention* model generates MEIs for which their Gaussian envelope on average is smaller than for the *Gaussian* MEIs ($\sigma_{At}$ = 49.62 vs $\sigma_{Ga}$ = 55.36, Wilcoxon signed rank test p-value: 0.0078).
> ### RE Q1: Increase in energy scale results in lower performance
> Thank you for pointing this out. The reason is that under the formulation $\varepsilon =  \varepsilon(x_t, t) + \lambda \nabla_x (x_t)$ increasing the $\lambda$ parameter is related to the step size between each diffusion step. Thus, for very large $\lambda$ the step becomes too big and thus resulting in lower performance. We cannot put $\lambda$ on $\varepsilon(x_t, t)$ as this results in the noise level that is out-of-distribution for the diffusion model and thus the diffusion process would fail.
> ### RE Q2: Other regularization techniques
> See general comment: **Re 4: Additional Related Work**
> ### RE Q3: Can we gain understanding from having things that “look” better?
> If enhanced visual representations (MEIs) were solely about aesthetics, they might be deceptive. However, EGG-generated MEIs not only look better, but they also enhance predicted neural responses and exhibit better generalization across architectures. Importantly, their improved visual quality aligns with improved performance, making their enhanced appearance a valuable aspect of interpretability.
> ### RE Q4: Biases on clean images
> We consider the proposed method as a way to understand what image features neurons are encoding. Since the visual system has evolved for natural images many experiments use them as stimuli. Thus also the encoding models are trained on these “clean” images. Our method needs to be able to deal with this and generate images that drive the modeled neurons well. For that reason, we use the clean sample prediction trick to circumvent the requirement for an encoding model trained on noisy images. This could potentially introduce some bias, but it’s hard to quantify it, in particular since we don’t have neuronal responses to noisy images.
> ### RE Q5: Spread across seeds
> Spread between seeds in Macaque V4 is expected [1] due to single-cell invariances. To show this we attach a figure of the MEIs from different seeds (Fig. A) (limited examples for the rebuttal due to space limitation, we will include more in the revised version of the manuscript).
> ### RE Clr a) Pretraining details for ResNet50 and ensemble
> The pretrained network is the L2, $\epsilon = 0.1$ ResNet50 obtained from [2]. For the ensemble model, we use the same core but separately trained readouts (same architecture, different weights).
> ### RE Clr b) 1024-dimensional feature space
> The ResNet50 at layer 3 has 1024 channels, as the readout selects a single point in that layer and performs a dot product between the readout weight vector and the 1024 channels. Therefore, this results in a 1024-dimensional feature space.
> ### RE Clr c) Missing Citation
> We will include the citation for "Solving linear inverse problems using the prior explicit in a denoiser"
> ### Re Clr d) Additional line for $\hat{x}_0$
> We will include the equation for computing the predicted $x_0$
> ### Re Clr e) Units activation space
> Yes, the unit activation space is the same as the predicted neural response. To avoid confusion between the predicted neural responses (model outputs) and real neural responses, we refer to predicted neural responses as unit responses.
> ### RE Typos/Confusions
> i) constant value of lambda - For generating all of the MEIs we use a single value of lambda, meaning that we do not select the best lambda for each neuron. We choose the overall lambda based on the validation model as shown in Figure 5. The goal is to simulate a paradigm where we can identify select lambda within our model, but the end goal is to optimize the best MEI for the brain (not the model itself)
>
> ii) $\epsilon_\theta$  typo - that is correct, thank you for spotting this typo, we will fix it in the revised manuscript.
> ### RE: Limitations
> - Not tested in the brain - that is a good point, we will make sure to discuss it in the limitations section.
> - Negative societal impact - We have thought about the negative societal impact, however, this is a method for fundamental neuroscience research. All the scenarios we were able to come up with seemed far-fetched. If you have a particular scenario in mind, we are happy to discuss it.
>
> **References**
>
> [1] Willeke et al. “Deep learning-driven characterization of single cell tuning in primate visual area V4 unveils topological organization” (2023)
>
> [2] Salman et al. “Do Adversarially Robust ImageNet Models Transfer Better?” (2020)

---

> > ### Comment · Reviewer_YaDz · 2023-08-14
> >
> > Thank you for providing these clarifications and updated results! I've read through the responses to other reviewers, looked at the updated PDF, and went back through the related parts of the paper.
> >
> > The ablation study is a nice contribution to clarify that the attention readout is the more important part of the model. I also appreciate the discussion about "naturalness" and the various experiments the authors have performed to quantify their claims. I've updated my score based on this.
> >
> > One concern that I still have, is that the optimization that is occurring and the use of constraints may lead to the diffusion to "fail". This seems to be the general idea of the discussion with Reviewer H4Az, and I share similar concerns about whether the different models are tuned correctly to result in a fair comparison (ie how increasing the energy scale eventually decreases the performance). I think the authors should think deeply about this, and, in addition to including some of the references provided by H4Az, also list such concerns as a limitation. Nevertheless, I think that the overall idea is a nice contribution to the line of work generating MEIs.

---

> > > ### Author Response · Authors · 2023-08-18
> > > **Response to Reviewer YaDz**
> > >
> > > Thank you for increasing your score. We are happy to address your remaining concerns.
> > >
> > > ### RE: Backbones for ablation study
> > > You are correct, we use the Robust ResNet as the backbone for the Task-Driven setup and the CNN for the Data-Driven setup. We did not consider alternative architectures because the CNN has been picked from previous work that optimized the architecture to be fitted to neuronal data directly and the ResNet was designed to be trained on large-scale image datasets like ImageNet. To show this we additionally conducted a test where we trained the ResNet model directly on the neuronal data, like the CNN. In this case, the model achieves performance of 0.25 test correlation with the Gaussian Readout and 0.26 with the Attention readout, which is not as good as when using the Data-Driven core. We will make sure the backbones are clear when adding the Ablation study to the manuscript.
> > >
> > > ### RE: Optimization failure
> > > We agree that the case of increasing the energy scale which eventually decreases the performance is a limitation of the way energy guidance operates. We agree that it is important for this to be clear and will make sure to discuss it as a limitation. However, one should note that while the performance decreases at higher lambdas, it still performs better than GA (Fig. G, where 1 on the y-axis refers to the activation achieved by GA MEIs).  Since increasing the energy scale brings the generation process closer to GA, this decrease is this expected.

---

### Author Rebuttal · Authors · 2023-08-08

We appreciate the thorough and constructive reviews and are glad to see that the reviewers found our work to be **ingenious** (**GN1s**), **novel** (**YaDz**, **GN1s**, **H4Az**, **BnXF**), **well-written** (**H4Az**) and **clear** (**GN1s**, **H4Az**). We are happy to see that Reviewer **YaDz** **enjoyed** the paper and is **excited** by our method.

However, the reviewers also raised some concerns including:
1. It is not possible to determine which change in the proposed model leads to better predictivity (**YaDz**, **GN1s**)
2. Request for additional quantitative evaluations (**YaDz**, **H4Az**)
3. Can the method be applied to other neural experimental techniques like two-photon, single-photon microscopy? (**BnXF**)
4. More comprehensive discussion of prior work (**YaDz**, **H4Az**)

We discuss the above points here, and any remaining points in the individual comments below. We are confident that we can address these concerns and are happy to clarify any remaining issues in the discussion. In the comments, we refer to the figures in the uploaded PDF document.

### Re 1: Ablation Study
We performed an ablation study comparing the effects of transitioning from the *Task-Driven ResNet + Gaussian Readout* model to the *Data-Driven CNN + Attention Readout* model as requested by Reviewers **YaDz**, **GN1s**. We measure performance in terms of test correlation.

| Core \ Readout | Gaussian        | Attention         |
| -------------------- | ------------------- | ------------------- |
| Task-Driven      | 0.262                 | 0.276 (+5%)   |
| Data-Driven      | 0.229 (-13%)     | 0.294 (+12%) |

The percentages in the parentheses denote the change in performance relative to the *Task-Driven model with Gaussian readout*. This shows that shifting to the *attention readout* improves the performance for both *Task-Driven* and *Data-Driven* cores.

### Re 2: Quantitative Evaluations
**YaDz** asked for experimental proof that the attention readout does indeed use its ability to shift receptive fields based on the input image. We show this by inspecting the attention mask of the attention model and computing the average distance between the center of mass of the upper 5% percentile of this mask across different images for each neuron. We plot this against the test correlation of each neuron observing that the attention readout does perform shifts (Fig. E). We also include qualitative examples of the masks and the means in Fig. E.

**H4Az** suggested characterizing the distributional similarity of the EGG-generated most exciting natural inputs and the ImageNet top-k natural images using standard image metrics like Fréchet inception distance (FID). We measured the FID score between the generated images at different $\lambda$ values and the top-5 ImageNet images (Fig. D). Our results show that by changing $\lambda$ we approach the natural images manifold (lower FID).

Furthermore **H4Az** suggested using SSIM and VGG perceptual loss to additionally evaluate the performance of reconstructing images from predicted neural responses. We did not include that previously because, as shown in [1], metrics like SSIM are not necessarily a good predictor of how well neuronal responses are reproduced *in vivo*. Similarly, we observe that SSIM and VGG perceptual loss show no improvement (Fig. D). However, to strengthen our claim on the improvement of our reconstructed images we conducted a two-alternative forced choice task with 45 voluntary participants on 50 test images (Fig. C). The participants were instructed to choose which image (GD optimized or EGG generated) was more similar to the ground truth image. Results show an 82.22% average preference for EGG-generated images (95% confidence interval [80.59%, 83.75%]; Wilson score interval).

### Re 3: Applicability to other Neural Experimental Techniques
Reviewer **BnXF** asked whether our method can be used with other experimental techniques than electrophysiology. While the dataset we use for this study was recorded from in macaque visual cortex, it is in principle possible to use EGG for MEI generation and reconstructions with calcium imaging similar to the GA method on two-photon data in [2]. In fact, EGG can be applied to any modality that yields an encoding model. Showing this, however, on other experimental techniques is out of scope for this paper and one week of response time.

### Re 4: Additional Related Work
Thank you for pointing out additional references. We will include and discuss the following references in the revised manuscript.

- Nichol, Alex, et al. "Glide: Towards photorealistic image generation and editing with text-guided diffusion models." arXiv preprint arXiv:2112.10741 (2021).
- Ramesh, Aditya, et al. "Hierarchical text-conditional image generation with clip latents." arXiv preprint arXiv:2204.06125 (2022).
- crowsonkb's open-source work: https://github.com/afiaka87/clip-guided-diffusion
- Li, Wei, et al. "UPainting: Unified Text-to-Image Diffusion Generation with Cross-modal Guidance." arXiv preprint arXiv:2210.16031 (2022).
- Bashivan, Kar, DiCarlo "Neural population control via deepimage synthesis" (2019)
- Ponce et al. "Evolving Images for Visual Neurons Using a Deep Generative Network Reveals Coding Principles and Neuronal Preferences" (2019) and follow-up work
- Engstrom et al. "Adversarial Robustness as a Prior for Learned Representations"(2019)
- Feather et al. "Model metamers illuminate divergences between biological and artificial neural networks" (2022)
- Kadkhodaie & Simoncelli "Solving Linear Inverse Problems Using the Prior Implicit in a Denoiser" (2020)

**References**

[1] Cobos et al. “It takes neurons to understand neurons: Digital twins of visual cortex synthesize neural metamers” (2022)

[2] Walker et al. “Inception loops discover what excites neurons most using deep predictive models” (2019)

---

> ### Comment · Reviewer_YaDz · 2023-08-14
> **quick clarification regarding ablation experiment**
>
> Thanks to the authors for providing these responses! A quick clarification -- in the ablation experiment what is the backbone for the models? Is "Task-Driven" always a ResNet and is "Data-Driven" always a CNN? If so, is there a reason for this? Its a little unclear to me from the table.

---

> > ### Author Response · Authors · 2023-08-18
> > **Response to Reviewer YaDz**
> >
> > Responded in detail in the thread with reviewer YaDz. To summarize: we use the Robust ResNet as the backbone for the Task-Driven setup and the CNN for the Data-Driven setup.

---

### Decision · Program_Chairs · 2023-09-21

**Decision:**

Accept (poster)

**Comment:**

There is a general consensus that the contribution of the paper is novel. The reviewers initially raised a number of comments regarding the experimental design and the interpretation/presentation of the results. The rebuttal appeared to address the majority of these concerns after a lengthy back-and-forth between the authors and the reviewers. Overall, there was moderate but unanimous support for the paper. The AC thus recommends the paper be accepted.